# Explaining Similarity in Vision–Language Encoders with Weighted Banzhaf Interactions

**Hubert Baniecki**[*]
University of Warsaw
Warsaw University of Technology

**Maximilian Muschalik**
LMU Munich, MCML

**Fabian Fumagalli**
Bielefeld University

**Barbara Hammer**
Bielefeld University

**Eyke Hüllermeier**
LMU Munich, MCML, DFKI

**Przemyslaw Biecek**
University of Warsaw
Warsaw University of Technology

## Abstract

Language–image pre-training (LIP) enables the development of vision–language models capable of zero-shot classification, localization, multimodal retrieval, and semantic understanding. Various explanation methods have been proposed to visualize the importance of input image–text pairs on the model's similarity outputs. However, popular saliency maps are limited by capturing only first-order attributions, overlooking the complex cross-modal interactions intrinsic to such encoders. We introduce faithful interaction explanations of LIP models (FIxLIP) as a unified approach to decomposing the similarity in vision–language encoders. FIxLIP is rooted in game theory, where we analyze how using the weighted Banzhaf interaction index offers greater flexibility and improves computational efficiency over the Shapley interaction quantification framework. From a practical perspective, we propose how to naturally extend explanation evaluation metrics, such as the pointing game and area between the insertion/deletion curves, to second-order interaction explanations. Experiments on the MS COCO and ImageNet-1k benchmarks validate that second-order methods, such as FIxLIP, outperform first-order attribution methods. Beyond delivering high-quality explanations, we demonstrate the utility of FIxLIP in comparing different models, e.g. CLIP vs. SigLIP-2.

## 1 Introduction

Contrastive language–image pre-training [CLIP, 57] revolutionized computer vision by learning data representations well-performing in a plethora of downstream tasks. Scaling the training to predict similarity between image and text, combined with continuous architectural improvements [e.g. SigLIP, 75] and increasing the size of datasets, leads to the development of powerful vision–language encoders [VLEs, 13, 66, 68, 75]. These rather opaque encoders become the backbone components in large vision–language models capable of actual conversation and reasoning [2, 36, 43], and are increasingly applied in high-stakes decision-making, like in the case of medical imaging [29, 76, 78].

The limitations in the way CLIP represents the visual world have been extensively studied [14, 35, 67, 72, 73]. For example, a recent study by Tong et al. [63] suggests vision–language models are limited by the systematic shortcomings of CLIP, e.g. they fail when questioned about directions, quantity of objects in an image, or recognizing the presented text in a visual form. The emergence of "CLIP-blind pairs", inputs that CLIP perceives as similar despite their clear differences, could lead to critical errors when applied in medical visual question answering and treatment recommendation [78].

---

[*]Corresponding author: h.baniecki@uw.edu.pl    Code: https://github.com/hbaniecki/fixlip

39th Conference on Neural Information Processing Systems (NeurIPS 2025).

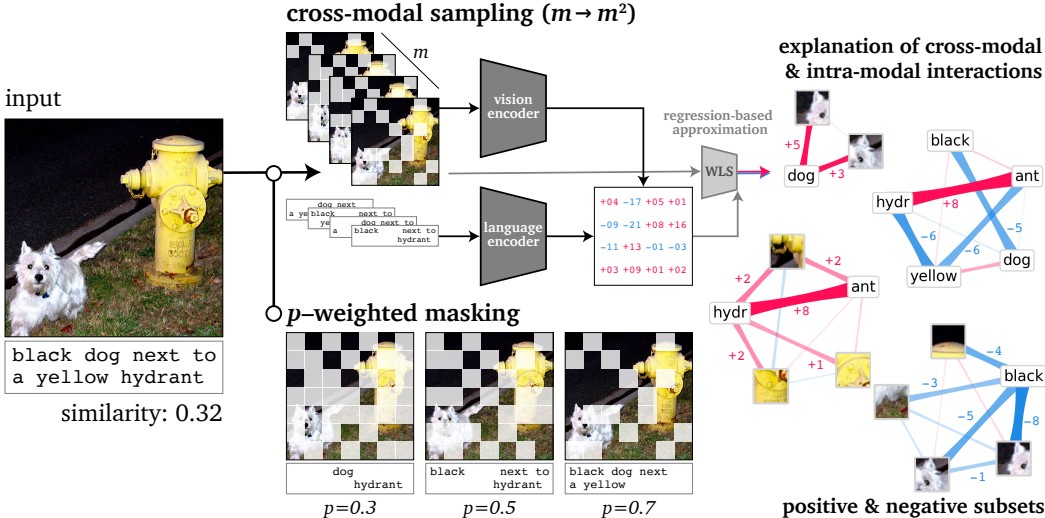

**Figure 1: Explaining similarity in vision–language encoders with weighted Banzhaf interactions.** We propose a cross-modal sampling strategy to efficiently query the model for $m^2$ game values from $m$ coalitions, and $p$-weighted masking to circumvent querying the model on out-of-distribution inputs. A regression-based approximation with weighted least squares (WLS) of second-order attributions gives a faithful decomposition of the predicted similarity score. Explanations of cross-modal and intra–modal interactions can be visualized and analyzed to interpret the CLIP's similarity prediction. The red values denote positive interactions contributing to an increase in similarity, while blue denotes interactions between tokens contributing to a decrease in similarity.

One way of validating similarity predictions at inference time is explaining the encoders with saliency maps [12, 39, 49, 70, 77]. However, these first-order importance scores are limited by their unimodal view of the model's output [1]. Vision–language encoders cannot be faithfully explained without taking into account the cross-modal interactions between image–text pairs [32, 40].

We thus propose a model-agnostic approach to interpreting VLEs based on a game-theoretical perspective where input tokens form coalitions of players in a cooperative game [19, 44]. In doing that, we extend the faithful Banzhaf interaction index [64] to a weighted case [47], addressing the emerging requirement for controllable sparsity in coalition sampling [6, 25, 27, 79]. Specifically, sparse inputs with many masked tokens become out-of-distribution, i.e. images become unrecognizable and text captions become ambiguous. Figure 1 shows a high-level view of our approach. We sample uniformly $m^2$ variations of the input image–text pair. Each mask is sampled uniformly with probability $p$ for each token in a mask, and we take all pairwise combinations between $m$ masked images and $m$ masked texts. Then, we apply the explained model to predict similarity values for each of the input variations. Finally, we apply weighted least squares to approximate the model's predictions from sampled binary masks. The resulting explanation assigns attribution scores to each token and interaction scores to each pair of tokens in an image–text pair.

**Contributions.** Our work advances literature in multiple ways: **(1) A game-theoretic explanation of vision–language encoders.** We introduce faithful interaction explanations of LIP models (FIxLIP) that provide a unique perspective on decomposing the similarity predictions of VLEs. Our results highlight the *necessity* to consider cross-modal interactions between text and image inputs for an accurate model interpretation, beyond the first-order attribution. In fact, this is the first work to argue for using *weighted* Banzhaf interactions in this domain to overcome the out-of-distribution problem apparent in removal-based explainability. **(2) Efficient computation allows scaling to larger models.** We propose a *cross-modal* sampling strategy for approximating FIxLIP that improves its efficiency by 5–20× over traditional Shapley interaction quantification. We further scale the regression-based approximation of interactions to hundreds of tokens by prioritizing their subset for a faithful explanation. **(3) Evaluation metrics for second-order interactions.** We extend explanation evaluation metrics—the pointing game and area between the insertion/deletion curves—to explanation methods of higher order. Experiments on the MS COCO and ImageNet-1k benchmarks validate that second-order methods, such as FIxLIP, outperform first-order attributions. **(4) FIxLIP facilitates**

**various approaches to understanding vision–language encoders.** Finally, we demonstrate the utility of FIxLIP in applications like a model-agnostic comparison between CLIP and SigLIP-2.

## 2    Related Work

**Attribution explanations of CLIP.** Multimodal explanations have been studied in various tasks, including visual question answering [28, 55], visual reasoning [46], video classification [71], and sentiment analysis [69]. Our work considers explaining the bi-modal encoder prediction in the means of input attributions, specifically image and text token attributions of similarity predictions in CLIP. In this context, existing gradient-based [77], attention-based [12, 39], and information-based [70, 80] methods approximate only *first-order* attributions visualized as saliency maps. Yet, Liang et al. [40] perform extensive user studies showing that visualizing the *second-order* attributions (pairwise interactions) is necessary for understanding complex multimodal models. Joukovsky et al. [32] propose to approximate these interactions with bilinear models [also for image–image encoders, 61]. We formalize the problem in the means of game theory and propose an efficient approximation of cross-modal and intra-modal interactions (Figure 1), discussing its several desirable theoretical properties. In concurrent work, Moeller et al. [48] propose to approximate only cross-modal interactions based on the Integrated Hessians methodology [30]. Our proposed evaluation metrics for interaction-based explanations can be used to empirically cross-compare the above-mentioned attribution explanations.

**Mechanistic interpretability of CLIP.** Our research is orthogonal to work on the concept-based interpretability of CLIP's internal representations. Research in this direction considers explaining particular neurons [22], concept-based image classification [60], linear probing [5], and training sparse autoencoders [41, 74]. Gandelsman et al. [21] investigate how the individual model's components affect its final representation. Beyond CLIP, Balasubramanian et al. [4] analyze interpreting image representations via text in alternative vision transformer architectures [e.g. DINO, 11].

**Shapley and Banzhaf interactions in machine learning.** We build on the developments using game-theoretic interaction indices to understand complex machine learning models [9, 19, 56, 62, 64]. Our goal in this work is not to compare with the most recent advancements in approximating the interaction index [16, 33, 52], but rather to faithfully explain the similarity predicted by VLEs. From the first-order perspective, Parcalabescu and Frank [53, 54] apply the Shapley value to measure the importance each modality has on various tasks solved by CLIP. Tsai et al. [64] and Fumagalli et al. [20] propose a weighted least squares regression approximation of higher-order interactions, which we incorporate as a baseline in our methodology. Recently, Kang et al. [34] applied a sparse Fourier transform to scale feature interaction explanations for large language models, while we explain vision–language encoders instead. Related to our contribution of weighted Banzhaf interaction explanations is work on using weighted Banzhaf values to estimate data valuation scores [37, 38], which are the importance that training data points or subsets have on the model's performance. Jin et al. [31] use Banzhaf interactions to enhance the training of video–language encoders. For a comprehensive overview of work in this direction, see [50, 59] and references given there.

## 3    A Game-theoretic Explanation of Similarity in Vision–Language Encoders

A vision–language encoder $f$ consists of a vision encoder $f_{\mathcal{I}} : \mathbb{R}^{n_{\mathcal{I}}} \to \mathbb{R}^d$ with $n_{\mathcal{I}} \in \mathbb{N}$ image patches (tokens), and a language encoder $f_{\mathcal{T}} : \mathbb{R}^{n_{\mathcal{T}}} \to \mathbb{R}^d$ with $n_{\mathcal{T}} \in \mathbb{N}$ text tokens. For a given input pair $(x_{\mathcal{I}}, x_{\mathcal{T}})$, the model computes a cosine similarity of their $d$-dimensional embeddings as

$$f(x_{\mathcal{I}}, x_{\mathcal{T}}) \coloneqq \cos\big(f_{\mathcal{I}}(x_{\mathcal{I}}), f_{\mathcal{T}}(x_{\mathcal{T}})\big) = \frac{f_{\mathcal{I}}(x_{\mathcal{I}}) \cdot f_{\mathcal{T}}(x_{\mathcal{T}})}{\|f_{\mathcal{I}}(x_{\mathcal{I}})\| \cdot \|f_{\mathcal{T}}(x_{\mathcal{T}})\|}. \tag{1}$$

In this context, we index the set of image tokens with $N_{\mathcal{I}} \coloneqq \{1, \dots, n_{\mathcal{I}}\}$ and the set of text tokens with $N_{\mathcal{T}} \coloneqq \{n_{\mathcal{I}} + 1, \dots, n_{\mathcal{I}} + n_{\mathcal{T}}\}$. Related attribution methods construct explanations that assign importance values to all individual input tokens. To understand cross-modal and intra-modal relationships between tokens, we extend these explanations by adding all pairwise token interactions.

**Definition 1** (Explanation). *An explanation* $\mathbf{e} \in \mathbb{R}^{|\mathcal{B}|}$ *assigns a constant* $\mathbf{e}_0$*, individual attributions* $\mathbf{e}_i \in \mathbb{R}$ *and pairwise interactions* $\mathbf{e}_{\{i,j\}} \in \mathbb{R}$*, for an explanation basis*

$$\mathcal{B} \coloneqq \underbrace{\{0\}}_{constant} \cup \underbrace{\{i : i \in N_{\mathcal{I}} \cup N_{\mathcal{T}}\}}_{individual\ tokens} \cup \underbrace{\{\{i,j\} : i,j \in N_{\mathcal{I}} \cup N_{\mathcal{T}}, i \neq j\}}_{two\text{-}token\ interaction\ sets}.$$

An interaction explanation can be viewed as a complete graph with weighted nodes and edges, similar to the SI-Graph [51], or a symmetric quadratic matrix of dimension $n_\mathcal{I} + n_\mathcal{T}$, where the diagonal represents token attributions. To compute explanations, we mask tokens with a pre-defined baseline, depending on the encoder's architecture, e.g. attention masking for text tokens (we defer implementation details to Appendix B). The masking operator is used to define the FIxLIP game that captures all possible masks.

**Definition 2** (Masking). *We define the masking operator $\oplus_{M_o} : \mathbb{R}^n \times \mathbb{R}^n \to \mathbb{R}^n$ for $n \in \{n_\mathcal{I}, n_\mathcal{T}\}$ that compute for a subset of indices $M_o \subseteq N$ with $N \in \{N_\mathcal{I}, N_\mathcal{T}\}$ and inputs $x, b \in \mathbb{R}^n$ as*

$$x \oplus_{M_o} b := \begin{cases} x_i, & \text{if } i \in M_o, \\ b_i, & \text{if } i \in N \setminus M_o. \end{cases}$$

**Definition 3** (Game). *For an input image–text pair $(x_\mathcal{I}, x_\mathcal{T})$ and a baseline $(b_\mathcal{I}, b_\mathcal{T})$ indexed by $N_\mathcal{I} \cup N_\mathcal{T}$, the FIxLIP explanation game $\nu : 2^{N_\mathcal{I} \cup N_\mathcal{T}} \to \mathbb{R}$ measures the similarity of the inputs given any mask $M \subseteq N_\mathcal{I} \cup N_\mathcal{T}$ as $\nu(M) := \nu(M \cap N_\mathcal{I}, M \cap N_\mathcal{T}) := f(x_\mathcal{I} \oplus_{M \cap N_\mathcal{I}} b_\mathcal{I}, x_\mathcal{T} \oplus_{M \cap N_\mathcal{T}} b_\mathcal{T})$.*

The FIxLIP game measures the similarity of the masked image and text inputs for every possible mask. In the following, we will quantify attributions of individual tokens, as well as cross- and intra-modal interactions, that *faithfully* explain the FIxLIP game, and satisfy important axioms.

### 3.1 FIxLIP-$p$ explanations via weighted faithful Banzhaf interactions

In this section, we formally define the FIxLIP explanation as an instance of Definition 1, which faithfully describes the similarity of all masked inputs $\nu(M)$. We thus define $\mathbf{e}$ as an *additive explanation with interaction terms* that recovers $\nu(M) \approx \hat{\nu}_\mathbf{e}(M)$ via a 2-additive game [24] as

$$\hat{\nu}_\mathbf{e}(M) := \mathbf{e}_0 + \sum_{i \in M} \mathbf{e}_i + \sum_{\{i,j\} \subseteq M : i \neq j} \mathbf{e}_{\{i,j\}} \text{ for all masks } M \subseteq N_\mathcal{I} \cup N_\mathcal{T}. \tag{2}$$

A natural choice of faithfulness is to measure the squared error $\big(\nu(M) - \hat{\nu}_\mathbf{e}(M)\big)^2$ across all masks. To further distinguish in-distribution (few tokens masked) and out-of-distribution (many tokens masked) inputs, we associate a mask weight according to its size $|M|$ in the following metric.

**Definition 4** ($p$-faithfulness). *For $p \in (0,1)$ and $\nu, \hat{\nu} : 2^{N_\mathcal{I} \cup N_\mathcal{T}} \to \mathbb{R}$, we measure $p$-faithfulness $\mathfrak{F}_p$ as $\mathfrak{F}_p(\nu, \hat{\nu}) := \sum_{M \subseteq N_\mathcal{I} \cup N_\mathcal{T}} p^{|M|}(1-p)^{n_\mathcal{I} + n_\mathcal{T} - |M|}\big(\nu(M) - \hat{\nu}(M)\big)^2$.*

**Remark 1.** *Computing $p$-faithfulness requires evaluating $2^{n_\mathcal{I} + n_\mathcal{T}}$ masks, which is infeasible in practice. Instead, we write $\mathfrak{F}_p$ as an expectation over randomly generated masks $\mathfrak{F}_p = \mathbb{E}_{M \sim \mathbb{P}_p}\big[\big(\nu(M) - \hat{\nu}(M)\big)^2\big]$, where $\mathbb{P}_p(M) := p^{|M|}(1-p)^{n_\mathcal{I} + n_\mathcal{T} - |M|}$ is a probability distribution over $2^{N_\mathcal{I} \cup N_\mathcal{T}}$. This formulation allows us to estimate $\mathfrak{F}_p$ using Monte Carlo integration.*

The hyperparameter $p$ determines the importance of each mask and can be viewed as the probability of a token being active, i.e. its index being included in $M$. The case $p = 0.5$ equally weights all masks, $p > 0.5$ prioritizes masks with many active tokens, i.e. preferring in-distribution inputs, whereas $p < 0.5$ prioritizes sparse masks preferring more out-of-distribution inputs.

**Definition 5** (FIxLIP-$p$). *We define the FIxLIP-$p$ explanation as $\mathbf{e}^{\text{FIxLIP-}p} := \text{argmin}_\mathbf{e} \, \mathfrak{F}_p(\nu, \hat{\nu}_\mathbf{e})$.*

The FIxLIP-$p$ explanation is the most faithful approximation of $\nu$ with respect to $\mathfrak{F}_p$. Its interactions correspond to the weighted Banzhaf interactions [47]. For $p = 0.5$, FIxLIP-$p$ is the faithful Banzhaf interaction index [64] with a known analytical solution [26].

**Remark 2.** *Weighted Banzhaf interactions define a cardinal-probabilistic interaction index that satisfies the linearity, symmetry, and dummy axiom [18]. Alternatively, faithful Shapley interactions optimize a variant of the faithfulness metric [64]. In our context, weighted Banzhaf interactions have two benefits: First and foremost, the hyperparameter $p$ allows for prioritizing masks that are expected to better reflect in-distribution inputs, and provides a flexible explanation framework with intuitive weights. Second, it is unknown if the Shapley weights can be factored into independent distributions for each modality – a crucial requirement for the cross-modal estimator introduced in Section 3.2.*

FIxLIP-$p$ explanations of VLEs offer a unique perspective on interpreting image–text similarity predictions. Notably, the resulting second-order interaction explanation can always be simplified into a first-order attribution explanation and visualized as a traditional saliency map over input tokens.

**Theorem 1** (First-order conversion). *For $i \in N_{\mathcal{I}} \cup N_{\mathcal{T}}$ and $\mathbf{e}^{\text{FIxLIP-}p}$, the first-order attribution values are given by $\mathbf{e}_i + p \sum_{j \in N_{\mathcal{I}} \cup N_{\mathcal{T}}: j \neq i} \mathbf{e}_{\{i,j\}}$, which are the weighted Banzhaf values of $\hat{\nu}$.*

We defer the proofs to Appendix A. While FIxLIP-$p$ can be computed analytically, it still requires $2^{n_{\mathcal{I}} + n_{\mathcal{T}}}$ mask evaluations. We thus introduce a model-agnostic estimator for FIxLIP-$p$ by estimating $\mathfrak{F}_p$ via Monte Carlo integration with sampled masks and optimizing the least-squares objective.

**Definition 6** (Model-agnostic estimator). *Let $M^{(1)}, \ldots, M^{(m)} \overset{iid}{\sim} \mathbb{P}_p$. The model-agnostic estimator is then $\hat{\mathbf{e}}^{\text{FIxLIP-}p} \coloneqq \arg\min_{\mathbf{e}} \hat{\mathfrak{F}}_p^{(m)}(\nu, \hat{\nu}_{\mathbf{e}})$, where $\hat{\mathfrak{F}}_p^{(m)}(\nu, \hat{\nu}_{\mathbf{e}}) \coloneqq \frac{1}{m} \sum_{\ell=1}^{m} \left( \nu(M^{(\ell)}) - \hat{\nu}_{\mathbf{e}}(M^{(\ell)}) \right)^2$.*

For $p = 0.5$, this estimator reduces to Faith-Banzhaf [64] – a variant of KernelSHAP [44].

### 3.2 Cross-modal sampling with $p$-weighted masking

We observe that the model-agnostic estimator uses naive sampling over masks $M \sim \mathbb{P}_p$, which becomes prohibitive for models with a large number of input tokens. Since every token is independently active with probability $p$, we propose to separately sample $m_{\mathcal{I}}$ subsets for the image and $m_{\mathcal{T}}$ subsets for the text modality, i.e. to decompose $\mathbb{P}_p = \mathbb{P}_{p,\mathcal{I}} \otimes \mathbb{P}_{p,\mathcal{T}}$, and sample independently. This allows for generating all possible combinations, obtaining a novel estimator of $\mathfrak{F}_p$.

**Definition 7** (Cross-modal estimator). *Let $M_{\mathcal{I}}^{(1)}, \ldots, M_{\mathcal{I}}^{(m_{\mathcal{I}})} \overset{iid}{\sim} \mathbb{P}_{p,\mathcal{I}}$ and $M_{\mathcal{T}}^{(1)}, \ldots, M_{\mathcal{T}}^{(m_{\mathcal{T}})} \overset{iid}{\sim} \mathbb{P}_{p,\mathcal{T}}$. We define the cross-modal estimator as $\hat{\mathbf{e}}^{\text{FIxLIP-}p} \coloneqq \arg\min_{\mathbf{e}} \hat{\mathfrak{F}}_p^{(m_{\mathcal{I}}, m_{\mathcal{T}})}(\nu, \hat{\nu}_{\mathbf{e}})$, where*

$$\hat{\mathfrak{F}}_p^{(m_{\mathcal{I}}, m_{\mathcal{T}})}(\nu, \hat{\nu}_{\mathbf{e}}) \coloneqq \frac{1}{m_{\mathcal{I}} \cdot m_{\mathcal{T}}} \sum_{\ell_{\mathcal{I}}=1}^{m_{\mathcal{I}}} \sum_{\ell_{\mathcal{T}}=1}^{m_{\mathcal{T}}} \left( \nu(M_{\mathcal{I}}^{(\ell_{\mathcal{I}})} \cup M_{\mathcal{T}}^{(\ell_{\mathcal{T}})}) - \hat{\nu}_{\mathbf{e}}(M_{\mathcal{I}}^{(\ell_{\mathcal{I}})} \cup M_{\mathcal{T}}^{(\ell_{\mathcal{T}})}) \right)^2.$$

Crucially, $\hat{\mathfrak{F}}_p^{(m_{\mathcal{I}}, m_{\mathcal{T}})}$ can be computed efficiently because $f$ independently encodes images for $m_{\mathcal{I}}$ samples with $f_{\mathcal{I}}$ and texts for $m_{\mathcal{T}}$ samples with $f_{\mathcal{T}}$ (cf. Equation 1). Then, similarity predictions are computed in a vectorized manner between all the cross-modal combinations of masked inputs. In theory, given $m = m_{\mathcal{I}} + m_{\mathcal{T}}$ samples, the model-agnostic estimator obtains $m$ game values (similarity predictions), while our proposed cross-modal estimator obtains $m_{\mathcal{I}} \cdot m_{\mathcal{T}} \gg m$. The empirically observed computational speedup, which we analyze in Section 5.4, can depend on the model's batch size (closely related to the GPU VRAM hardware). Note that reusing the samples in text and image modalities inflicts dependence between them, and the following result establishes important theoretical properties.

**Theorem 2.** *The cross-modal estimator $\hat{\mathfrak{F}}_p^{(m_{\mathcal{I}}, m_{\mathcal{T}})}$ is unbiased, and its variance is bounded by*

$$\mathbb{V}\left[ \hat{\mathfrak{F}}_p^{(m_{\mathcal{I}} \cdot m_{\mathcal{T}})}(\nu, \hat{\nu}_{\mathbf{e}}) \right] \leq \mathbb{V}\left[ \hat{\mathfrak{F}}_p^{(m_{\mathcal{I}}, m_{\mathcal{T}})}(\nu, \hat{\nu}_{\mathbf{e}}) \right] \leq \frac{m_{\mathcal{I}} + m_{\mathcal{T}}}{m_{\mathcal{I}} \cdot m_{\mathcal{T}}} \mathbb{V}\left[ \left( \nu(M_{\mathcal{I}} \cup M_{\mathcal{T}}) - \hat{\nu}_{\mathbf{e}}(M_{\mathcal{I}} \cup M_{\mathcal{T}}) \right)^2 \right].$$

In other words, Theorem 2 shows that the variance of the cross-modal estimator is at most of the same order as the variance of the model-agnostic estimator with $m \approx m_{\mathcal{I}} \approx m_{\mathcal{T}}$ samples. Moreover, the variance is at least the variance of the model-agnostic estimator with $m_{\mathcal{I}} \cdot m_{\mathcal{T}}$ independent samples. In practice, we use the cross-modal estimator in higher budget scenarios to speed up the computation by decreasing the number of effective model inferences.

### 3.3 Large-scale adaptations for FIxLIP explanations

In contrast to first-order explanations, the explanation basis of Definition 1 grows quadratically as $|\mathcal{B}| = 1 + n_{\mathcal{I}} + n_{\mathcal{T}} + \binom{n_{\mathcal{I}} + n_{\mathcal{T}}}{2} \approx (n_{\mathcal{I}} + n_{\mathcal{T}})^2$. For example, a ViT-B/16 version of CLIP with 196 image and 30 text input tokens (players) results already in 25 425 interactions. From a practical perspective, we thus consider two heuristics as part of our complete methodology: **(1)** We employ a *two-step filtering approach* for prioritizing interactions to approximate when explaining games with a large number of players. A default is to pick a clique subset with the highest (absolute) first-order attributions and compute interactions between them, e.g. top-72 with $\binom{72}{2} = 2\,556$ interactions. Another approach is to include only cross-modal interactions in the approximation, e.g. $196 \times 30 = 5\,880$. **(2)** We apply a simple *greedy subset selection* algorithm on explanation $\mathbf{e}$ to find subgraphs $M \subseteq N_{\mathcal{I}} \cup N_{\mathcal{T}}$ of the highest and lowest sums of $\hat{\nu}_{\mathbf{e}}(M)$, which we use in evaluating (Section 5.1) and visualizing (Section 5.5) explanations. Further details are in Appendix B.

# 4 Evaluation Metrics for Interaction Explanations of VLE Predictions

In this section, we derive three evaluation metrics for explanations that may include second-order interactions. First, we evaluate the faithfulness of the approximation $\hat{\nu}_{\mathbf{e}}$ to $\nu$, since the basis of explaining VLEs is the masking operator that defines the FIxLIP game $\nu$. Note that conventionally this is done with an $R^2$ coefficient [33, 50] or mean squared error [16, 19, 52], but we need to rely on rankings to compare attribution methods like saliency maps. Second, we generalize the area between insertion and deletion curves [AID, 25, 77], aka remove least/most important first [6], a popular faithfulness metric for token attribution methods. Finally, we extend the well-established pointing game evaluation of faithfulness from the image classification literature [3, 7, 8, 48]. In the latter two metrics, we need to assume $\mathbf{e}_{\{i,j\}} = 0$ for the first-order attribution explanations without interactions.

We validate the faithfulness to the FIxLIP game $\nu$ across sampled subsets. To this end, we compare the rankings of $\nu$ and $\hat{\nu}$, since related explanation methods are not normalized to estimate $\nu$ correctly.

**Definition 8** ($p$-faithfulness correlation)**.** *We define $p$-faithfulness correlation as the Spearman's rank correlation computed between $\nu$ and $\hat{\nu}_{\mathbf{e}}$ evaluated for $m$ masks sampled from $\mathbb{P}_p$ (cf. Remark 1) as*

$$\mu_{\mathrm{corr}}(\mathbf{e}; p) := \underset{M^{(1)},\dots,M^{(m)} \overset{\mathrm{iid}}{\sim} \mathbb{P}_p}{\mathrm{correlation}} \left( \nu(M^{(i)}), \hat{\nu}_{\mathbf{e}}(M^{(i)}) \right).$$

The rank correlation metric $\mu_{\mathrm{corr}}$ yields insights into the general capability of recovering the ranking of similarity scores of uniformly masked inputs based on $\mathbb{P}_p$.

Beyond capturing the general $p$-faithfulness, we derive an insertion/deletion visualization with a metric measuring the explanation's ability to identify token subsets (masks) of high and low similarity as evaluated by the game (model).

**Definition 9** (Area between insertion/deletion curves (AID))**.** *For each size $k = 1, \dots, n_{\mathcal{I}} + n_{\mathcal{T}}$, let $M_{\mathbf{e},\min,k} := \mathrm{argmin}_{M \subseteq N_{\mathcal{I}} \cup N_{\mathcal{T}}: |M|=k} \hat{\nu}_{\mathbf{e}}(M)$ and $M_{\mathbf{e},\max,k} := \mathrm{argmax}_{M \subseteq N_{\mathcal{I}} \cup N_{\mathcal{T}}: |M|=k} \hat{\nu}_{\mathbf{e}}(M)$. Insertion and deletion curves are computed as $\mathcal{C}_{\mathrm{insert}}(\mathbf{e}) := \left\{ \left( k, \nu(M_{\mathbf{e},\max,k}) \right) \right\}_{k=1,\dots,n_{\mathcal{I}}+n_{\mathcal{T}}}$, and $\mathcal{C}_{\mathrm{delete}}(\mathbf{e}) := \left\{ \left( k, \nu(M_{\mathbf{e},\min,n_{\mathcal{I}}+n_{\mathcal{T}}-k}) \right) \right\}_{k=1,\dots,n_{\mathcal{I}}+n_{\mathcal{T}}}$, respectively. The metric is then defined as*

$$\mu_{\mathrm{AID}}(\mathbf{e}) := \int \mathcal{C}_{\mathrm{insert}}(\mathbf{e}) - \int \mathcal{C}_{\mathrm{delete}}(\mathbf{e}) = \sum_{k=1}^{n_{\mathcal{I}}+n_{\mathcal{T}}} \left( \nu(M_{\mathbf{e},\max,k}) - \nu(M_{\mathbf{e},\min,k}) \right).$$

$\mu_{\mathrm{AID}}$ measures the difference between model predictions when including and excluding the most relevant tokens. For first-order methods, $M_{\mathbf{e},\min,k}$ and $M_{\mathbf{e},\max,k}$ are directly found by ranking $\mathbf{e}$.

**Proposition 1.** *For token attribution values, i.e. when $\forall_{i,j}\ \mathbf{e}_{\{i,j\}} \equiv 0$, such as weighted Banzhaf values, $M_{\mathbf{e},\max,k}$ and $M_{\mathbf{e},\min,k}$ are given by the top-$k$ and bottom-$k$ coefficients of $\mathbf{e}$, respectively. Consequently, $\mathcal{C}_{\mathrm{delete}}$ is found by deleting $M_{\mathbf{e},\max,k}$, i.e. $M_{\mathbf{e},\min,n_{\mathcal{I}}+n_{\mathcal{T}}-k} = (N_{\mathcal{I}} \cup N_{\mathcal{T}}) \setminus M_{\mathbf{e},\max,k}$.*

Consequently, $\mathcal{C}_{\mathrm{insert}}$ and $\mathcal{C}_{\mathrm{delete}}$ generalize insertion and deletion curves to second-order interaction explanations. Note that for token attribution values, we have $M_{\mathbf{e},\max,k} \subset M_{\mathbf{e},\max,k+1}$, and $M_{\mathbf{e},\min,k} \subset M_{\mathbf{e},\min,k+1}$, which does not hold when modeling interaction terms. We give a further description of the insertion/deletion process on a single image–text input in Appendix B.2, Figure 9.

Lastly, we propose to evaluate an explanation $\mathbf{e}$ with designed pseudo ground-truth data. We measure the overlap between an explanation of inputs crafted as four combined images with a multi-object text prompt, and its "ground-truth" assumed prior. We show a visual example in Appendix B.3, Figure 8

**Definition 10** (Pointing game recognition (PGR))**.** *For $\mathbf{e}$, let $\mathbf{e}_{\mathrm{in},k}$ and $\mathbf{e}_{\mathrm{out},k}$ denote the values of interactions belonging to, and not belonging to, object $k$ – a text token with its corresponding image patches. We denote the positive and negative interactions by $\mathbf{e}_{>0}, \mathbf{e}_{<0}$. The metric is then defined as*

$$\mu_{\mathrm{PGR}}(\mathbf{e}; N_{\mathcal{T}}) := \frac{\sum_{k \in N_{\mathcal{T}}} \left( \|\mathbf{e}_{\mathrm{in},k,>0}\|_1 + \|\mathbf{e}_{\mathrm{out},k,<0}\|_1 \right)}{\sum_{k \in N_{\mathcal{T}}} \left( \|\mathbf{e}_{\mathrm{in},k}\|_1 + \|\mathbf{e}_{\mathrm{out},k}\|_1 \right)} \in [0,1], \text{ where } \|\mathbf{e}\|_1 := \sum_{i,j} |\mathbf{e}_{\{i,j\}}|.$$

$\mu_{\mathrm{PGR}}$ quantifies the ratio of absolute values of "correctly" identified cross-modal interaction terms to total interaction terms. Thereby, "correctly" identified cross-modal interactions refer to positive and negative interactions that are associated with and without object $k$ in an image, respectively. We use $\mu_{\mathrm{PGR}}$ as a sanity check that interaction explanations are essential for explaining VLEs as compared to alternative attribution approaches. Scoring high PGR values across multiple objects present in an image–text pair denotes that an explanation method can show the model distinguishes between them.

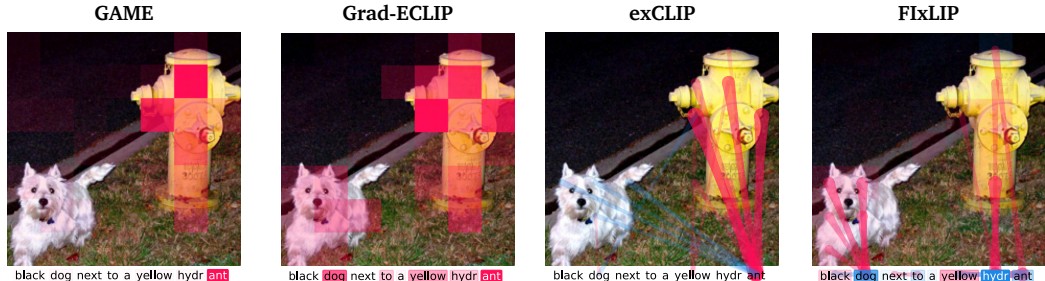

**Figure 2: Visual comparison between FIxLIP and baselines.** First-order attribution methods, e.g. GAME and Grad-ECLIP, lack the tools to faithfully explain complex similarity predictions of vision–language encoders like CLIP. Notably, in this example, the text token ant 🐜 is the most important for the similarity prediction. One of the differences from exCLIP is that we include intra-modal and main effects in the approximation, which are crucial for obtaining faithful interaction explanations.

## 5 Experiments

**Setup.** In experiments, we empirically validate the performance of FIxLIP with three metrics defined in Section 4, measure its computational efficiency, and demonstrate its utility in visual explanation of VLEs. We mainly use the openly available pre-trained CLIP models [57] of two sizes: ViT-B/32 with $7 \times 7$ image patches and ViT-B/16 with $14 \times 14$. Moreover, we demonstrate the broader applicability of FIxLIP to explain SigLIP [75] and SigLIP-2 [66] up to the ViT-L/16 variant with $16 \times 16$ patches. We rely on two openly available datasets commonly used in explainability research: MS COCO [42] and ImageNet-1k [15]; the latter specifically to design the pointing game evaluation considering zero-shot classification.

In quantitative evaluation, we compare FIxLIP to a few representative baselines: the first attribution method for CLIP abbreviated as GAME [12], a state-of-the-art first-order attribution method Grad-ECLIP [77], and a recently released second-order interaction method exCLIP [48]. For comparisons of Grad-ECLIP and exCLIP with previous baselines, refer to the appropriate benchmarks [48, 77]. Figure 2 conceptually compares FIxLIP to baselines using the image–text input example from Figure 1. Furthermore, we naturally demonstrate the improvement between FIxLIP and first-order Shapley/Banzhaf values. In that, we effectively scale FIxLIP to approximate second-order explanations with a budget of over $10^6$ model inferences, which far exceeds related work. For FIxLIP-$p$, we use the cross-modal estimator with a budget of $2^{21}$, whereas FIxLIP with Shapley interactions uses the model-agnostic estimator with a budget of $2^{17}$, yielding approximately similar runtime. We mainly experiment with $p \in \{0.3, 0.5, 0.7\}$; note that there is no computational overhead for different $p$ settings. Further details for the experimental setup are provided in Appendix C.

### 5.1 FIxLIP outperforms baselines as measured with insertion/deletion curves

Figure 3 shows insertion/deletion results for different explanation methods, where each line and metric value is an average ($\pm$ sd.) over 1000 inputs. We observe that FIxLIP faithfully recovers the nonlinear importance rankings of token subsets (see Appendix Figure 9 for a visual example). It not only finds the most important tokens whose deletion results in a significant drop in similarity (y-axis), but also **finds the least important tokens whose deletion may even result in the prediction's increase**, contrary to gradient-based methods. The general conclusion is consistent for SigLIP-2 and a larger ViT-B/16 model (refer to Appendix D for additional results), albeit the method's faithfulness drops when increasing the size of input tokens, where further scaling to even higher image resolutions is a natural future work direction. As designed, increasing the masking weight to $p = 0.7$ leads to an improved faithfulness in the 0–50% range of input deleted (100–50% inserted), while decreasing to $p = 0.3$ improves faithfulness with 50–100% input deleted (50–0% inserted). Note that, in theory, all of the methods can obtain negative normalized values on the y-axis. We think that exCLIP fails to recover the appropriate ranking because it only approximates cross-modal interactions between tokens from the two modalities, omitting first-order effects and intra-modal interactions in the process (effectively constructing a bipartite weighted graph without weights in nodes).

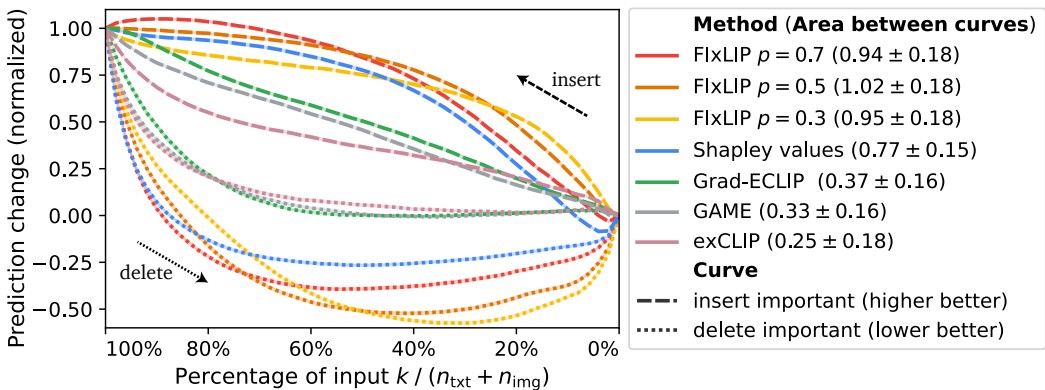

**Figure 3: Insertion/deletion curves for CLIP (ViT-B/32) on MS COCO.** AID score (higher is better) for FIxLIP against alternative explanation methods, where a random baseline scores $0$. The y-axis is normalized between the model's prediction on the original input ($100\%$) and the fully removed one ($0\%$), where negative values denote that the model is predicting the image–text inputs are unsimilar. It means the similarity prediction on a partially masked input is smaller than the prediction on the fully masked input. Methods such as Grad-ECLIP and exCLIP fail to recover nonlinear rankings of important tokens, while our method faithfully recovers the optimal subset explanation. Extended results for CLIP (ViT-B/16) and SigLIP-2 (ViT-B/32) are in Figures 10 & 11.

**Table 1: Pointing game recognition for CLIP (ViT-B/32) on ImageNet-1k.** PGR score (higher is better) for FIxLIP against alternative explanation methods, where a random baseline scores $0.25$. First-order methods, such as Grad-ECLIP and Shapley values, fail to distinguish between multiple objects at once, while second-order methods faithfully recover the appropriate explanation (up to the pointing game's irreducible non-optimality). Extended results for CLIP (ViT-B/16) are in Table 4.

| Explanation Method | Recognition ($\uparrow$) | | | |
| --- | --- | --- | --- | --- |
| | 1 object | 2 objects | 3 objects | 4 objects |
| GAME [12] | $.61_{\pm.12}$ | $.43_{\pm.03}$ | $.33_{\pm.02}$ | $.28_{\pm.01}$ |
| Grad-ECLIP [77] | $.68_{\pm.15}$ | $.45_{\pm.04}$ | $.33_{\pm.02}$ | $.28_{\pm.01}$ |
| Shapley values | $.70_{\pm.11}$ | $.56_{\pm.06}$ | $.46_{\pm.05}$ | $.37_{\pm.04}$ |
| Banzhaf values | $.64_{\pm.12}$ | $.52_{\pm.06}$ | $.43_{\pm.05}$ | $.35_{\pm.04}$ |
| exCLIP [48] | $.73_{\pm.20}$ | $.88_{\pm.08}$ | $.89_{\pm.06}$ | $.92_{\pm.05}$ |
| FIxLIP (Shapley interactions) | $.83_{\pm.10}$ | $.82_{\pm.08}$ | $.84_{\pm.06}$ | $.86_{\pm.06}$ |
| FIxLIP (w. Banzhaf interactions $p = 0.3$) | $.78_{\pm.13}$ | $.78_{\pm.11}$ | $.80_{\pm.08}$ | $.81_{\pm.07}$ |
| FIxLIP (w. Banzhaf interactions $p = 0.5$) | $.81_{\pm.12}$ | $.80_{\pm.09}$ | $.81_{\pm.07}$ | $.83_{\pm.06}$ |
| FIxLIP (w. Banzhaf interactions $p = 0.7$) | $.83_{\pm.12}$ | $.81_{\pm.08}$ | $.83_{\pm.07}$ | $.85_{\pm.06}$ |

## 5.2 First-order attribution methods fail to pass a sanity check with a pointing game

Table 1 shows PGR results for different explanation methods, where each metric value is an average ($\pm$ sd.) over 500 inputs. We observe that first-order methods, e.g. Banzhaf values, fail to discriminate between multiple objects in an image. Saliency maps can only highlight the important part of image–text inputs without disentangling the complex relationship between each separate word in a caption and the corresponding image regions (Appendix Figure 8). Second-order methods such as FIxLIP and exCLIP pass our proposed sanity check; the latter scores slightly higher, as it specializes in approximating cross-modal interactions. The conclusion is consistent for CLIP (ViT-B/16). We further analyze the potential application of a pointing game to evaluate different models in Section 5.6.

## 5.3 FIxLIP recovers a faithful decomposition of the similarity function

Figure 4 demonstrates $p$-faithfulness results for different explanation methods, where each boxplot represents statistics aggregated from 1000 inputs. We confirm that FIxLIP optimizes faithfulness, which depends on the parameter $p$. Weighted Banzhaf interactions allow for precise controllability of the faithfulness optimization, whereas Shapley interactions perform well on average. Alternative

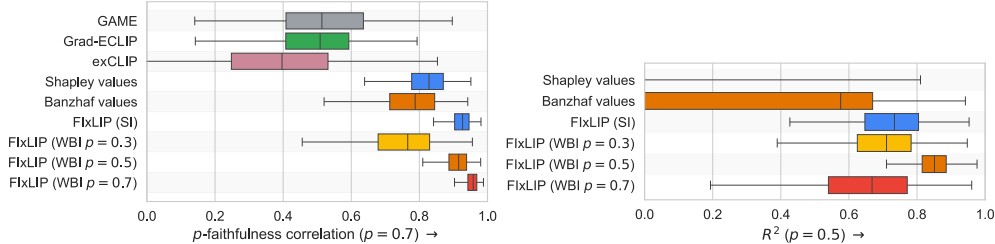

**Figure 4:** $p$-**faithfulness correlation for CLIP (ViT-B/32) on MS COCO.** Correlation for different variants of FIxLIP against other explanation methods (**left**). Game-theoretical approaches can also be evaluated with the $R^2$ coefficient (**right**). Extended results for CLIP (ViT-B/16) are in Figure 12.

attribution methods compute explanations unfaithful to the explained similarity function ($\mu_{\text{corr}} \approx 0.5$). Extended results for different models and $p$ values are given in Appendix D.

### 5.4 On the computational efficiency of the FIxLIP cross-modal estimator

Figure 5 demonstrates that the FIxLIP cross-modal estimator achieves over $20\times$ speedup when considering model inference time, i.e. game evaluations, and about $5\times$ speedup when accounting for the entire explanation pipeline implemented in Python. We did not necessarily optimize the latter, i.e. subset sampling, weighted least squares optimization, and other processing steps, expecting that the visible gap between the explanation and game time could be further decreased. For context, related first-order attribution methods take about 1 second to compute; see Table 5 in [77], but note that it could be only for an image explanation without text attribution.

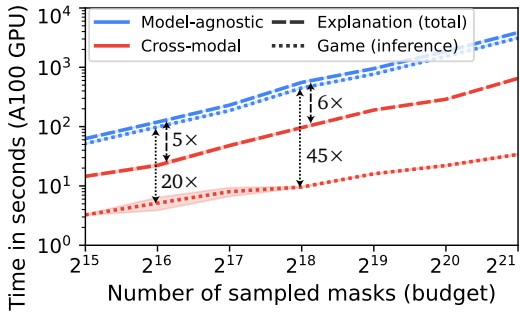

**Figure 5:** Computation time vs. budget for the FIxLIP explanation of SigLIP-2 (ViT-B/32), including game evaluations (model inference).

### 5.5 Visual explanation of vision–language encoders

We have already established that FIxLIP delivers state-of-the-art faithfulness performance across three diverse metrics. One of the explanations' applications is to interpret the model's output function. Figure 6 demonstrates three types of visualizations that we envision can be used to broaden our understanding of VLEs. In the interest of space, we provide further interesting examples in Appendix E. Specifically, we provide a comprehensive guide on interpreting interaction explanations like FIxLIP in Appendix E.1. Figures 8 & 9 further compare FIxLIP to baselines in the context of evaluation metrics. Figure 16 shows a comparison between FIxLIP of CLIP and SigLIP-2 (ViT-B/32), while Figure 17 shows a comparison between the ViT-B/32 and ViT-B/16 versions of CLIP.

### 5.6 Model-agnostic comparison of different vision–language encoder architectures

We use FIxLIP to compare different VLE architectures with PGR in Table 2. **Surprisingly, SigLIP-2 can be more faithfully explained with cross-modal interactions than CLIP**, for both model sizes. We observe that, in general, smaller models (ViT-B/32) can be more faithfully explained than larger ones (ViT-L/16), which is consistent with prior work [9, 16]. For more visual comparisons, see Appendix E.3.

**Table 2:** Pointing game results as measured with FIxLIP.

| Size | Model | Recognition ($\uparrow$) | | | |
|------|-------|----------|-----------|--------|--------|
| | | 1 object | 2 objects | 3 obj. | 4 obj. |
| ViT-B/32 | CLIP | $.83_{\pm.11}$ | $.81_{\pm.07}$ | $.82_{\pm.07}$ | $.85_{\pm.05}$ |
| | SigLIP-2 | $.90_{\pm.07}$ | $.90_{\pm.05}$ | $.89_{\pm.05}$ | $.89_{\pm.04}$ |
| ViT-B/16 | CLIP | $.81_{\pm.09}$ | $.81_{\pm.06}$ | $.81_{\pm.05}$ | $.82_{\pm.04}$ |
| | SigLIP | $.80_{\pm.08}$ | $.82_{\pm.06}$ | $.84_{\pm.05}$ | $.84_{\pm.05}$ |
| | SigLIP-2 | $.86_{\pm.07}$ | $.88_{\pm.04}$ | $.87_{\pm.04}$ | $.87_{\pm.03}$ |
| ViT-L/16 | SigLIP | $.81_{\pm.08}$ | $.84_{\pm.05}$ | $.85_{\pm.04}$ | $.84_{\pm.04}$ |
| | SigLIP-2 | $.80_{\pm.10}$ | $.85_{\pm.06}$ | $.84_{\pm.05}$ | $.85_{\pm.04}$ |

**(A) Faithful interaction explanations**          **(B) Conditioning on tokens**

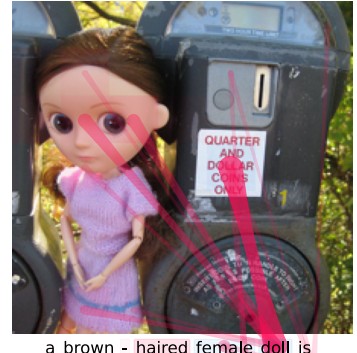

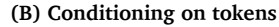

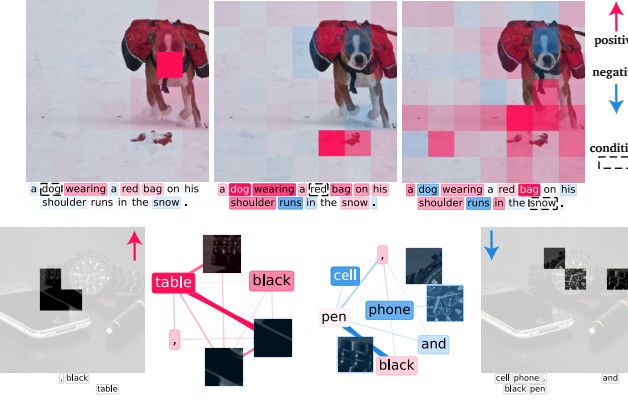

**(C) Visualizing subsets**

Figure 6: **FIxLIP facilitates various approaches to model understanding. (A)** Interaction explanations allow answering the pivotal question: *Why is it similar for the model?* Here, the strongest interaction is between text token `doll` and image patch saying "dollar". One could say the model is right for the wrong reasons. **(B)** Each token can be selected for conditioning to visualize as a heatmap only the interactions (edges) outgoing from it. **(C)** A complete graph can be traversed to find subsets of high positive and low negative similarity as approximated by the explanation.

## 6   Discussion

As vision–language encoders are increasingly deployed in real-world applications, it becomes pivotal to ensure that their predictions are explainable. To this end, we introduced faithful interaction explanations of CLIP and SigLIP models, offering a unique perspective on interpreting image–text similarity predictions. Moreover, we derived three evaluation criteria facilitating future work in this direction.

**Limitations and future work.** Our work faces three limitations, each with a clear path for future development. Although FIxLIP allows scaling to larger models with hundreds of players and efficiently computing a million model inferences, improvements could be made to its practical implementation. Specifically, we envision exploring the use of sparse linear regression, applying the Archipelago framework [65] to filter interactions for the approximation, and a non-greedy algorithm for a closer-to-optimal subset selection. Second, the visual properties and usability of FIxLIP should be further studied as a potential direction in human-computer interaction research [58], see e.g. a user study of cross-modal interaction explanations by Liang et al. [40]. Finally, our work is restricted to second-order interaction explanations, while efficiently approximating higher-order interactions (also in tri-modal settings beyond LIP models) becomes an interesting challenge to overcome.

**Broader impact.** We believe FIxLIP can empower model developers in debugging VLEs, understanding their similarity predictions, and finding unwanted biases in image–text data. Especially when these models are used in high-stakes decision making, like in the case of medical applications.

**Code.** We provide additional details on reproducibility in the Appendix, as well as the code to reproduce all experiments in this paper is available at https://github.com/hbaniecki/fixlip.

## Acknowledgments and Disclosure of Funding

We gratefully acknowledge the Polish high-performance computing infrastructure PLGrid (HPC Centers: ACK Cyfronet AGH) for providing computer facilities and support within the computational grant no. PLG/2025/018330. Hubert Baniecki was supported from the state budget within the Polish Ministry of Education and Science program "Pearls of Science" project number PN/01/0087/2022. Barbara Hammer, Eyke Hüllermeier, Fabian Fumagalli, and Maximilian Muschalik gratefully acknowledge funding by the Deutsche Forschungsgemeinschaft (DFG, German Research Foundation): TRR 318/1 2021 – 438445824.

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

# Appendix

Table 3 summarises the mathematical notation used in the paper. In Appendix A, we derive proofs for Theorems 1 & 2, and Proposition 1. Appendix B describes the implementation details of our methods introduced in Sections 3 & 4. Appendix C provides further details on the setup of experiments conducted in Section 5. Finally, we discuss ablation results in Appendix D and visualize additional explanations in Appendix E.

**Table 3:** Summary of notation.

| Notation | Description |
|---|---|
| $n_\mathcal{I} \in \mathbb{N}, n_\mathcal{T} \in \mathbb{N}$ | Image and text input dimensions (number of tokens) |
| $x_\mathcal{I} \in \mathbb{R}^{n_\mathcal{I}}, x_\mathcal{T} \in \mathbb{R}^{n_\mathcal{T}}$ | Image and text input vectors |
| $f(x_\mathcal{I}, x_\mathcal{T})$ | Vision–language encoder prediction for an image–text input pair |
| $d \in \mathbb{N}$ | Encoder embedding dimension |
| $f_\mathcal{I}(x_\mathcal{I}) \in \mathbb{R}^d, f_\mathcal{T}(x_\mathcal{T}) \in \mathbb{R}^d$ | Vision and language encoder embedding vectors |
| $N_\mathcal{I}, N_\mathcal{T}$ | Sets of image and text token indices |
| $i, j$ | Token indices |
| $\mathcal{B}$ | Explanation basis set containing indices of tokens and token pairs |
| $\mathbf{e} \in \mathbb{R}^{|\mathcal{B}|}$ | Explanation object |
| $\mathbf{e}_i, \mathbf{e}_{\{i,j\}} \in \mathbb{R}$ | Token attribution and interaction values |
| $M_o \subseteq N$ with $N \in \{N_\mathcal{I}, N_\mathcal{T}\}$ | Subset of image or text token indices |
| $M \subseteq N_\mathcal{I} \cup N_\mathcal{T}$ | Subset of image and text token indices |
| $b_\mathcal{I}, b_\mathcal{T}$ | Baseline image and text vectors |
| $x \oplus_{M_o} b$ | Input $x$ masked using the operator for subset $M_o$ and baseline $b$ |
| $\nu(M)$ | Game (value) function |
| $\hat{\nu}, \hat{\nu}_\mathbf{e}$ | Estimators of the game (value) function |
| $p \in (0, 1)$ | Probability parameter |
| $\mathfrak{F}_p(\nu, \hat{\nu})$ | $p$-faithfulness error metric |
| $\mathbf{e}^{\text{FIXLIP-}p} := \arg\min_\mathbf{e} \mathfrak{F}_p(\nu, \hat{\nu}_\mathbf{e})$ | Explanation approximating the game function w.r.t. $p$-faithfulness |
| $\mathbb{P}_p(M) := p^{|M|}(1-p)^{n_\mathcal{I}+n_\mathcal{T}-|M|}$ | Probability distribution of sampled masks over $2^{N_\mathcal{I} \cup N_\mathcal{T}}$ |
| $m$ | Number of sampled mask subsets |
| $M^{(1)}, \ldots, M^{(m)}$ | Sampled mask subsets of image and text token indices |
| $\hat{\mathfrak{F}}_p^{(m)}(\nu, \hat{\nu}_\mathbf{e})$ | Model-agnostic estimator of $p$-faithfulness |
| $\mathbb{P}_{p,\mathcal{I}} \ \mathbb{P}_{p,\mathcal{T}}$ | Probability distributions of sampled masks over $2^{N_\mathcal{I}}$ and $2^{N_\mathcal{T}}$ |
| $m_\mathcal{I}, m_\mathcal{T}$ | Numbers of sampled image and text mask subsets |
| $M^{(1)}, \ldots, M^{(m_\mathcal{I})}$ | Sampled mask subsets of image token indices |
| $M^{(1)}, \ldots, M^{(m_\mathcal{T})}$ | Sampled mask subsets of text token indices |
| $\hat{\mathfrak{F}}_p^{(m_\mathcal{I}, m_\mathcal{T})}(\nu, \hat{\nu}_\mathbf{e})$ | Cross-modal estimator of $p$-faithfulness |
| $\mu_{\text{corr}}(\mathbf{e}; p)$ | $p$-faithfulness Spearman's rank correlation evaluation metric |
| $\mu_{\text{AID}}(\mathbf{e})$ | Area between insertion/deletion curves evaluation metric |
| $M_{\mathbf{e},\min,k}$ | Mask subset of size $k$ for explanation $\mathbf{e}$ minimizing $\hat{\nu}_\mathbf{e}(M_{\mathbf{e},\min,k})$ |
| $M_{\mathbf{e},\max,k}$ | Mask subset of size $k$ for explanation $\mathbf{e}$ maximizing $\hat{\nu}_\mathbf{e}(M_{\mathbf{e},\min,k})$ |
| $C_{\text{insert}}(\mathbf{e})$ | Insertion curve; a set of paired $(k, \nu(M_{\mathbf{e},\max,k}))$ values |
| $C_{\text{delete}}(\mathbf{e})$ | Deletion curve; a set of paired $(k, \nu(M_{\mathbf{e},\min,n_\mathcal{I}+n_\mathcal{T}-k}))$ values |
| $\mu_{\text{PGR}}(\mathbf{e}; N_\mathcal{T})$ | Pointing game recognition evaluation metric |
| $\mathbf{e}_{\text{in},k}$ | Interaction values belonging to text token $k$ |
| $\mathbf{e}_{\text{out},k}$ | Interaction values *not* belonging to text token $k$ |
| $\mathbf{e}_{>0}, \mathbf{e}_{<0}$ | Negative and positive interaction values in an explanation |
| $\|\mathbf{e}\|_1 := \sum_{i,j} |\mathbf{e}_{\{i,j\}}|$ | Absolute sum of an explanation |

# A  Proofs

## A.1  Proof of Theorem 1

*Proof.* To prove this result, we combine alternative representations of the cooperative games using the Möbius transform $a : 2^{N_\mathcal{I} \cup N_\mathcal{T}} \to \mathbb{R}$, which is defined by

$$a(M) := \sum_{L \subseteq M} (-1)^{|M|-|L|} \nu(L) \text{ for every } M \subseteq N_\mathcal{I} \cup N_\mathcal{T}.$$

The approximated game $\hat{\nu}_\mathbf{e}$ is a 2-additive game, and hence its Möbius transform is restricted up to second-order interactions [24]. Moreover, the Möbius transform of the approximated game $\hat{a}^{\text{FIXLIP}}$ satisfies

$$\hat{a}^{\text{FIXLIP-}p}(\emptyset) = \hat{\nu}_\mathbf{e}(\emptyset) = \mathbf{e}_0,$$
$$\hat{a}^{\text{FIXLIP-}p}(\{i\}) = \hat{\nu}_\mathbf{e}(\{i\}) - \hat{\nu}_\mathbf{e}(\emptyset) = \mathbf{e}_i,$$
$$\hat{a}^{\text{FIXLIP-}p}(\{i,j\}) = \hat{\nu}_\mathbf{e}(\{i,j\}) - \hat{\nu}_\mathbf{e}(\{i\}) - \hat{\nu}_\mathbf{e}(\{j\}) + \hat{\nu}_\mathbf{e}(\emptyset) = \mathbf{e}_{\{i,j\}},$$

where $i, j \in N_\mathcal{I} \cup N_\mathcal{T}$ and $i \neq j$. The approximated game $\hat{\nu}$ is by construction a 2-additive game, and thus all higher-order Möbius coefficients are zero [24]. It was further shown by Marichal and Mathonet [47, Proposition 3] that the weighted Banzhaf values of $\hat{\nu}$ can be computed as the optimal first-order approximation of $\hat{\nu}$. The representation of the best $k$-th order approximation in the Möbius transform was given by Marichal and Mathonet [47, Proposition 6], and reads for $i \in N_\mathcal{I} \cup N_\mathcal{T}$, $k = 1$, and equal probabilities $p$ as

$$\hat{a}(\{i\}) + (-1)^{1-1} \sum_{M \subseteq N_\mathcal{I} \cup N_\mathcal{T}:|M|>1, i \in M} \binom{|M|-1-1}{1-1} p^{|M|-1} \hat{a}(M)$$
$$= \hat{a}(\{i\}) + p \sum_{j \in N_\mathcal{I} \cup N_\mathcal{T}:j\neq i} \hat{a}(\{i,j\})$$
$$= \mathbf{e}_i + p \sum_{j \in N_\mathcal{I} \cup N_\mathcal{T}:j\neq i} \mathbf{e}_{\{i,j\}},$$

which concludes the proof. □

## A.2 Proof of Theorem 2

*Proof.* We first compute the expectations as

$$\mathbb{E}[\hat{\mathfrak{F}}_p^{(m_\mathcal{I},m_\mathcal{T})}(\nu,\hat{\nu}_\mathbf{e})] = \mathbb{E}\left[\frac{1}{m_\mathcal{I} m_\mathcal{T}} \sum_{\ell_\mathcal{I}=1}^{m_\mathcal{I}} \sum_{\ell_\mathcal{T}=1}^{m_\mathcal{T}} (\nu(M_\mathcal{I}^{(\ell_\mathcal{I})} \cup M_\mathcal{T}^{(\ell_\mathcal{T})}) - \hat{\nu}(M_\mathcal{I}^{(\ell_\mathcal{I})} \cup M_\mathcal{T}^{(\ell_\mathcal{T})}))^2\right]$$

$$= \frac{1}{m_\mathcal{I} m_\mathcal{T}} \sum_{\ell_\mathcal{I}=1}^{m_\mathcal{I}} \sum_{\ell_\mathcal{T}=1}^{m_\mathcal{T}} \mathbb{E}\left[(\nu(M_\mathcal{I}^{(\ell_\mathcal{I})} \cup M_\mathcal{T}^{(\ell_\mathcal{T})}) - \hat{\nu}(M_\mathcal{I}^{(\ell_\mathcal{I})} \cup M_\mathcal{T}^{(\ell_\mathcal{T})}))^2\right]$$

$$= \mathbb{E}_{(M_\mathcal{I},M_\mathcal{T})\sim\mathbb{P}_{p,\mathcal{I}}\otimes\mathbb{P}_{p,\mathcal{T}}}\left[(\nu(M_\mathcal{I} \cup M_\mathcal{T}) - \hat{\nu}(M_\mathcal{I} \cup M_\mathcal{T}))^2\right]$$

$$= \mathbb{E}_{M\sim\mathbb{P}}\left[(\nu(M) - \hat{\nu}(M))^2\right]$$

$$= \mathfrak{F}_p(\nu,\hat{\nu}_\mathbf{e}),$$

since $M_\mathcal{I} \perp M_\mathcal{T}$ are independent.

We now proceed by computing the variance. We first introduce $g_\mathbf{e} : 2^{N_\mathcal{I}} \times 2^{N_\mathcal{T}} \to \mathbb{R}$ for $M_\mathcal{I} \subseteq N_\mathcal{I}$ and $M_\mathcal{T} \subseteq N_\mathcal{T}$ as

$$g_\mathbf{e}(M_\mathcal{I}, M_\mathcal{T}) := (\nu(M_\mathcal{I} \cup M_\mathcal{T}) - \hat{\nu}_\mathbf{e}(M_\mathcal{I} \cup M_\mathcal{T}))^2.$$

For the standard Monte Carlo estimator, we then obtain

$$\mathbb{V}[\hat{\mathfrak{F}}_p^{(m)}(\nu,\hat{\nu}_\mathbf{e})] = \frac{1}{m}\mathbb{V}[g_\mathbf{e}(M_\mathcal{I}, M_\mathcal{T})],$$

due to $m$ iid samples. For the cross-modal estimator, we first prove a result for the explicit form of the variance in Proposition 2.

**Proposition 2.** *We denote the expectation over the conditional variances as*

$$\tau_{\mathcal{I}|\mathcal{T}} := \mathbb{E}_{M_\mathcal{T}\sim\mathbb{P}_{p,\mathcal{T}}}\left[\mathbb{V}_{M_\mathcal{I}\sim\mathbb{P}_{p,\mathcal{I}}}[(\nu(M_\mathcal{I} \cup M_\mathcal{T}) - \hat{\nu}_\mathbf{e}(M_\mathcal{I} \cup M_\mathcal{T}))^2]\right],$$

$$\tau_{\mathcal{T}|\mathcal{I}} := \mathbb{E}_{M_\mathcal{I}\sim\mathbb{P}_{p,\mathcal{I}}}\left[\mathbb{V}_{M_\mathcal{T}\sim\mathbb{P}_{p,\mathcal{T}}}[(\nu(M_\mathcal{I} \cup M_\mathcal{T}) - \hat{\nu}_\mathbf{e}(M_\mathcal{I} \cup M_\mathcal{T}))^2]\right].$$

*The variance of the estimator $\hat{\mathfrak{F}}_p^{(m_\mathcal{I},m_\mathcal{T})}(\nu,\hat{\nu}_\mathbf{e})$ is then given by*

$$\mathbb{V}[\hat{\mathfrak{F}}_p^{(m_\mathcal{I},m_\mathcal{T})}(\nu,\hat{\nu}_\mathbf{e})] = \frac{m_\mathcal{I} + m_\mathcal{T} - 1}{m_\mathcal{I} m_\mathcal{T}}\mathbb{V}[g_\mathbf{e}(M_\mathcal{I}, M_\mathcal{T})] - \frac{m_\mathcal{I} - 1}{m_\mathcal{I} m_\mathcal{T}}\tau_{\mathcal{I}|\mathcal{T}} - \frac{m_\mathcal{T} - 1}{m_\mathcal{I} m_\mathcal{T}}\tau_{\mathcal{T}|\mathcal{I}}.$$

*Proof.* The variance is computed as

$$\mathbb{V}[\hat{\mathfrak{F}}_p^{(m_\mathcal{I},m_\mathcal{T})}(\nu,\hat{\nu}_\mathbf{e})] = \mathbb{V}\left[\frac{1}{m_\mathcal{I} m_\mathcal{T}} \sum_{\ell_\mathcal{I}=1}^{m_\mathcal{I}} \sum_{\ell_\mathcal{T}=1}^{m_\mathcal{T}} g_\mathbf{e}\left(M_\mathcal{I}^{(\ell_\mathcal{I})}, M_\mathcal{T}^{(\ell_\mathcal{T})}\right)\right]$$

$$= \frac{1}{m_\mathcal{I}^2 m_\mathcal{T}^2} \sum_{\ell_\mathcal{I}=1}^{m_\mathcal{I}} \sum_{\ell_\mathcal{T}=1}^{m_\mathcal{T}} \sum_{k_\mathcal{I}=1}^{m_\mathcal{I}} \sum_{k_\mathcal{T}=1}^{m_\mathcal{T}} \mathrm{cov}\left(g_\mathbf{e}(M_\mathcal{I}^{(\ell_\mathcal{I})}, M_\mathcal{T}^{(\ell_\mathcal{T})}), g_\mathbf{e}(M_\mathcal{I}^{(k_\mathcal{I})}, M_\mathcal{T}^{(k_\mathcal{T})})\right). \tag{3}$$

To compute the covariance, we distinguish four cases, namely

$$\mathrm{cov}(g_\mathbf{e}(M_\mathcal{I}^{(\ell_\mathcal{I})}, M_\mathcal{T}^{(\ell_\mathcal{T})}), g_\mathbf{e}(M_\mathcal{I}^{(k_\mathcal{I})}, M_\mathcal{T}^{(k_\mathcal{T})}))$$

$$= \begin{cases} \mathbb{V}[g_\mathbf{e}(M_\mathcal{I}, M_\mathcal{T})], & \text{if } \ell_\mathcal{I} = k_\mathcal{I}, \ell_\mathcal{T} = k_\mathcal{T}, \\ \mathrm{cov}(g_\mathbf{e}(M_\mathcal{I}, M_\mathcal{T}), g_\mathbf{e}(M_\mathcal{I}, M_\mathcal{T}')), & \text{if } \ell_\mathcal{I} = k_\mathcal{I}, \ell_\mathcal{T} \neq k_\mathcal{T}, \\ \mathrm{cov}(g_\mathbf{e}(M_\mathcal{I}, M_\mathcal{T}), g_\mathbf{e}(M_\mathcal{I}', M_\mathcal{T})), & \text{if } \ell_\mathcal{I} \neq k_\mathcal{I}, \ell_\mathcal{T} = k_\mathcal{T}, \\ 0, & \text{if } \ell_\mathcal{I} \neq k_\mathcal{I}, \ell_\mathcal{T} \neq k_\mathcal{T}, \end{cases}$$

where $M_\mathcal{I}, M_\mathcal{I}'\sim\mathbb{P}_{p,\mathcal{I}}$ and $M_\mathcal{T}, M_\mathcal{T}'\sim\mathbb{P}_{\mathcal{T},p}$. The covariances can be further computed as

$$\mathrm{cov}(g_\mathbf{e}(M_\mathcal{I}, M_\mathcal{T}), g_\mathbf{e}(M_\mathcal{I}, M_\mathcal{T}')) = \mathbb{E}[g_\mathbf{e}(M_\mathcal{I}, M_\mathcal{T})g_\mathbf{e}(M_\mathcal{I}, M_\mathcal{T}')] - \mathbb{E}[g_\mathbf{e}(M_\mathcal{I}, M_\mathcal{T})]\mathbb{E}[g_\mathbf{e}(M_\mathcal{I}, M_\mathcal{T}')]$$

$$= \mathbb{E}_{M_\mathcal{I}\sim\mathbb{P}_{p,\mathcal{I}}}\left[\mathbb{E}_{M_\mathcal{T}\sim\mathbb{P}_{p,\mathcal{T}}}[g_\mathbf{e}(M_\mathcal{I}, M_\mathcal{T})]\mathbb{E}_{M_\mathcal{T}'\sim\mathbb{P}_{p,\mathcal{T}}}[g_\mathbf{e}(M_\mathcal{I}, M_\mathcal{T}')]\right] - \mathbb{E}[g_\mathbf{e}(M_\mathcal{I}, M_\mathcal{T})]^2$$

$$= \mathbb{E}_{M_\mathcal{I}\sim\mathbb{P}_{p,\mathcal{I}}}\left[\mathbb{E}_{M_\mathcal{T}\sim\mathbb{P}_{p,\mathcal{T}}}[g_\mathbf{e}(M_\mathcal{I}, M_\mathcal{T})]^2\right] - \mathbb{E}_{M_\mathcal{I}\sim\mathbb{P}_{p,\mathcal{I}}}\left[\mathbb{E}_{M_\mathcal{T}\sim\mathbb{P}_{p,\mathcal{T}}}[g_\mathbf{e}(M_\mathcal{I}, M_\mathcal{T})]\right]^2$$

$$= \mathbb{V}_{M_\mathcal{I}\sim\mathbb{P}_{p,\mathcal{I}}}\left[\mathbb{E}_{M_\mathcal{T}\sim\mathbb{P}_{p,\mathcal{T}}}[g_\mathbf{e}(M_\mathcal{I}, M_\mathcal{T})]\right],$$

where we have used $M_{\mathcal{I}} \perp M_{\mathcal{T}}, M'_{\mathcal{T}}$ and $M_{\mathcal{T}} \perp M'_{\mathcal{T}}$. Similarly, we obtain

$$\mathrm{cov}\left(g_{\mathbf{e}}(M_{\mathcal{I}}, M_{\mathcal{T}}), g_{\mathbf{e}}(M'_{\mathcal{I}}, M_{\mathcal{T}})\right) = \mathbb{V}_{M_{\mathcal{T}} \sim \mathbb{P}_{p,\mathcal{T}}}[\mathbb{E}_{M_{\mathcal{I}} \sim \mathbb{P}_{p,\mathcal{I}}}[g_{\mathbf{e}}(M_{\mathcal{I}}, M_{\mathcal{T}})]].$$

Combining these results into Eq. (3), we obtain

$$\mathbb{V}[\hat{\mathfrak{F}}_p^{(m_{\mathcal{I}}, m_{\mathcal{T}})}(\nu, \hat{\nu}_{\mathbf{e}})] = \frac{1}{m_{\mathcal{I}}^2 m_{\mathcal{T}}^2} \sum_{\ell_{\mathcal{I}}, k_{\mathcal{I}}=1}^{m_{\mathcal{I}}} \sum_{\ell_{\mathcal{T}}, k_{\mathcal{T}}=1}^{m_{\mathcal{T}}} \mathrm{cov}\left(g_{\mathbf{e}}(M_{\mathcal{I}}^{(\ell_{\mathcal{I}})}, M_{\mathcal{T}}^{(\ell_{\mathcal{T}})}), g_{\mathbf{e}}(M_{\mathcal{I}}^{(k_{\mathcal{I}})}, M_{\mathcal{T}}^{(k_{\mathcal{T}})})\right)$$

$$= \frac{1}{m_{\mathcal{I}}^2 m_{\mathcal{T}}^2}\Bigg(\underbrace{\sum_{\ell_{\mathcal{I}}=1}^{m_{\mathcal{I}}} \sum_{\ell_{\mathcal{T}}=1}^{m_{\mathcal{T}}} \mathbb{V}[g_{\mathbf{e}}(M_{\mathcal{I}}, M_{\mathcal{T}})]}_{\text{case } \ell_{\mathcal{I}}=k_{\mathcal{I}}, \ell_{\mathcal{T}}=k_{\mathcal{T}}}$$

$$+ \underbrace{\sum_{\ell_{\mathcal{I}}=1}^{m_{\mathcal{I}}} \sum_{\substack{\ell_{\mathcal{T}}, k_{\mathcal{T}}=1 \\ \ell_{\mathcal{T}} \neq k_{\mathcal{T}}}}^{m_{\mathcal{T}}} \mathbb{V}_{M_{\mathcal{I}} \sim \mathbb{P}_{p,\mathcal{I}}}\left[\mathbb{E}_{M_{\mathcal{T}} \sim \mathbb{P}_{p,\mathcal{T}}}[g_{\mathbf{e}}(M_{\mathcal{I}}, M_{\mathcal{T}})]\right]}_{\text{case } \ell_{\mathcal{I}}=k_{\mathcal{I}}, \ell_{\mathcal{T}} \neq k_{\mathcal{T}}}$$

$$+ \underbrace{\sum_{\substack{\ell_{\mathcal{I}}, k_{\mathcal{I}}=1 \\ \ell_{\mathcal{I}} \neq k_{\mathcal{I}}}}^{m_{\mathcal{I}}} \sum_{\ell_{\mathcal{T}}=1}^{m_{\mathcal{T}}} \mathbb{V}_{M_{\mathcal{T}} \sim \mathbb{P}_{p,\mathcal{T}}}\left[\mathbb{E}_{M_{\mathcal{I}} \sim \mathbb{P}_{p,\mathcal{I}}}[g_{\mathbf{e}}(M_{\mathcal{I}}, M_{\mathcal{T}})]\right]}_{\text{case } \ell_{\mathcal{I}} \neq k_{\mathcal{I}}, \ell_{\mathcal{T}}=k_{\mathcal{T}}}\Bigg)$$

$$= \frac{m_{\mathcal{I}} m_{\mathcal{T}}}{m_{\mathcal{I}}^2 m_{\mathcal{T}}^2} \mathbb{V}[g_{\mathbf{e}}(M_{\mathcal{I}}, M_{\mathcal{T}})]$$

$$+ \frac{m_{\mathcal{I}} m_{\mathcal{T}}(m_{\mathcal{T}}-1)}{m_{\mathcal{I}}^2 m_{\mathcal{T}}^2} \mathbb{V}_{M_{\mathcal{I}} \sim \mathbb{P}_{p,\mathcal{I}}}\left[\mathbb{E}_{M_{\mathcal{T}} \sim \mathbb{P}_{p,\mathcal{T}}}[g_{\mathbf{e}}(M_{\mathcal{I}}, M_{\mathcal{T}})]\right]$$

$$+ \frac{m_{\mathcal{T}} m_{\mathcal{I}}(m_{\mathcal{I}}-1)}{m_{\mathcal{I}}^2 m_{\mathcal{T}}^2} \mathbb{V}_{M_{\mathcal{T}} \sim \mathbb{P}_{p,\mathcal{T}}}\left[\mathbb{E}_{M_{\mathcal{I}} \sim \mathbb{P}_{p,\mathcal{I}}}[g_{\mathbf{e}}(M_{\mathcal{I}}, M_{\mathcal{T}})]\right]$$

We now use the law of total variance and $M_{\mathcal{I}} \perp M_{\mathcal{T}}$ to rewrite

$$\mathbb{V}_{M_{\mathcal{I}} \sim \mathbb{P}_{p,\mathcal{I}}}\left[\mathbb{E}_{M_{\mathcal{T}} \sim \mathbb{P}_{p,\mathcal{T}}}[g_{\mathbf{e}}(M_{\mathcal{I}}, M_{\mathcal{T}})]\right] = \mathbb{V}[g_{\mathbf{e}}(M_{\mathcal{I}}, M_{\mathcal{T}})] - \mathbb{E}_{M_{\mathcal{I}} \sim \mathbb{P}_{p,\mathcal{I}}}[\mathbb{V}_{M_{\mathcal{T}} \sim \mathbb{P}_{p,\mathcal{T}}}[g_{\mathbf{e}}(M_{\mathcal{I}}, M_{\mathcal{T}})]],$$

and use a similar result for $M_{\mathcal{T}}$ to obtain

$$\mathbb{V}[\hat{\mathfrak{F}}_p^{(m_{\mathcal{I}}, m_{\mathcal{T}})}(\nu, \hat{\nu}_{\mathbf{e}})] = \frac{m_{\mathcal{I}} m_{\mathcal{T}} + m_{\mathcal{I}} m_{\mathcal{T}}(m_{\mathcal{T}}-1) + m_{\mathcal{T}} m_{\mathcal{I}}(m_{\mathcal{I}}-1)}{m_{\mathcal{I}}^2 m_{\mathcal{T}}^2} \mathbb{V}[g_{\mathbf{e}}(M_{\mathcal{I}}, M_{\mathcal{T}})]$$

$$- \frac{m_{\mathcal{T}}-1}{m_{\mathcal{I}} m_{\mathcal{T}}} \mathbb{E}_{M_{\mathcal{I}} \sim \mathbb{P}_{p,\mathcal{I}}}[\mathbb{V}_{M_{\mathcal{T}} \sim \mathbb{P}_{p,\mathcal{T}}}[g_{\mathbf{e}}(M_{\mathcal{I}}, M_{\mathcal{T}})]] - \frac{m_{\mathcal{I}}-1}{m_{\mathcal{I}} m_{\mathcal{T}}} \mathbb{E}_{M_{\mathcal{T}} \sim \mathbb{P}_{p,\mathcal{T}}}[\mathbb{V}_{M_{\mathcal{I}} \sim \mathbb{P}_{p,\mathcal{I}}}[g_{\mathbf{e}}(M_{\mathcal{I}}, M_{\mathcal{T}})]]$$

$$= \frac{m_{\mathcal{I}} + m_{\mathcal{T}} - 1}{m_{\mathcal{I}} m_{\mathcal{T}}} \mathbb{V}[g_{\mathbf{e}}(M_{\mathcal{I}}, M_{\mathcal{T}})]$$

$$- \frac{(m_{\mathcal{T}}-1)\mathbb{E}_{M_{\mathcal{I}} \sim \mathbb{P}_{p,\mathcal{I}}}[\mathbb{V}_{M_{\mathcal{T}} \sim \mathbb{P}_{p,\mathcal{T}}}[g_{\mathbf{e}}(M_{\mathcal{I}}, M_{\mathcal{T}})]] + (m_{\mathcal{I}}-1)\mathbb{E}_{M_{\mathcal{T}} \sim \mathbb{P}_{p,\mathcal{T}}}[\mathbb{V}_{M_{\mathcal{I}} \sim \mathbb{P}_{p,\mathcal{I}}}[g_{\mathbf{e}}(M_{\mathcal{I}}, M_{\mathcal{T}})]]}{m_{\mathcal{I}} m_{\mathcal{T}}}$$

$$= \frac{m_{\mathcal{I}} + m_{\mathcal{T}} - 1}{m_{\mathcal{I}} m_{\mathcal{T}}} \mathbb{V}[g_{\mathbf{e}}(M_{\mathcal{I}}, M_{\mathcal{T}})] - \frac{m_{\mathcal{I}}-1}{m_{\mathcal{I}} m_{\mathcal{T}}} \tau_{\mathcal{I}|\mathcal{T}} - \frac{m_{\mathcal{T}}-1}{m_{\mathcal{I}} m_{\mathcal{T}}} \tau_{\mathcal{T}|\mathcal{I}},$$

where we have used the definitions of $\tau_{\mathcal{I}|\mathcal{T}} := \mathbb{E}_{M_{\mathcal{T}} \sim \mathbb{P}_{p,\mathcal{T}}}[\mathbb{V}_{M_{\mathcal{I}} \sim \mathbb{P}_{p,\mathcal{I}}}[g_{\mathbf{e}}(M_{\mathcal{I}}, M_{\mathcal{T}})]$ and $\tau_{\mathcal{T}|\mathcal{I}} := \mathbb{E}_{M_{\mathcal{I}} \sim \mathbb{P}_{p,\mathcal{I}}}[\mathbb{V}_{M_{\mathcal{T}} \sim \mathbb{P}_{p,\mathcal{T}}}[g_{\mathbf{e}}(M_{\mathcal{I}}, M_{\mathcal{T}})]]$. This concludes the proof. $\qquad\square$

By using Proposition 2, we obtain

$$\mathbb{V}[\hat{\mathfrak{F}}_p^{(m_{\mathcal{I}}, m_{\mathcal{T}})}(\nu, \hat{\nu}_{\mathbf{e}})] \leq \frac{m_{\mathcal{I}} + m_{\mathcal{T}} - 1}{m_{\mathcal{I}} m_{\mathcal{T}}} \mathbb{V}[g_{\mathbf{e}}(M_{\mathcal{I}}, M_{\mathcal{T}})] \leq \frac{m_{\mathcal{I}} + m_{\mathcal{T}}}{m_{\mathcal{I}} m_{\mathcal{T}}} \mathbb{V}[g_{\mathbf{e}}(M_{\mathcal{I}}, M_{\mathcal{T}})],$$

since $\tau_{\mathcal{I}|\mathcal{T}}, \tau_{\mathcal{T}|\mathcal{I}} \geq 0$.

Moreover, by the law of total variance we have $\tau_{\mathcal{I}|\mathcal{T}}, \tau_{\mathcal{T}|\mathcal{T}} \leq \mathbb{V}[g_{\mathbf{e}}(M_{\mathcal{I}}, M_{\mathcal{T}})]$, which implies

$$
\begin{aligned}
\mathbb{V}[\hat{\mathfrak{F}}_p^{(m,m)}(\nu, \hat{\nu}_{\mathbf{e}})] &\geq \frac{2m-1}{m^2} \mathbb{V}[g_{\mathbf{e}}(M_{\mathcal{I}}, M_{\mathcal{T}})] - \frac{2(m-1)}{m^2} \mathbb{V}[g_{\mathbf{e}}(M_{\mathcal{I}}, M_{\mathcal{T}})] \\
&= \frac{1}{m^2} \mathbb{V}[g_{\mathbf{e}}(M_{\mathcal{I}}, M_{\mathcal{T}})] \\
&= \mathbb{V}[\mathfrak{F}_p^{(m^2)}(\nu, \hat{\nu}_{\mathbf{e}})],
\end{aligned}
$$

which finishes the proof. $\qquad\square$

## A.3 Proof of Proposition 1

*Proof.* For a token attribution explanation $\mathbf{e}$, the additive approximation reads as

$$\hat{\nu}_{\mathbf{e}}(M) = \mathbf{e}_0 + \sum_{i \in M} \mathbf{e}_i.$$

Denote $M_k^+$ the top-$k$ coefficients of $\mathbf{e}$, then for $M \subseteq N_{\mathcal{I}} \cup N_{\mathcal{T}}$ with $|M| = k$, we have

$$\sum_{i \in M} \mathbf{e}_i \leq \sum_{i \in M_k^+} \mathbf{e}_i,$$

and thus

$$M_{\mathbf{e},\max,k} = \underset{M \subseteq N_{\mathcal{I}} \cup N_{\mathcal{T}}:|M|=k}{\arg\max} \hat{\nu}_{\mathbf{e}}(M) = \underset{M \subseteq N_{\mathcal{I}} \cup N_{\mathcal{T}}:|M|=k}{\arg\max} \sum_{i \in M} \mathbf{e}_i = M_k^+.$$

Conversely, for the bottom-$k$ coefficients $M_k^-$, we have

$$\sum_{i \in M} \mathbf{e}_i \geq \sum_{i \in M_k^-} \mathbf{e}_i,$$

for every $M \subseteq N_{\mathcal{I}} \cup N_{\mathcal{T}}$ with $|M| = k$, and thus

$$M_{\mathbf{e},\min,k} = \underset{M \subseteq N_{\mathcal{I}} \cup N_{\mathcal{T}}:|M|=k}{\arg\min} \hat{\nu}_{\mathbf{e}}(M) = \underset{M \subseteq N_{\mathcal{I}} \cup N_{\mathcal{T}}:|M|=k}{\arg\min} \sum_{i \in M} \mathbf{e}_i = M_k^-.$$

Lastly, the top-$k$ coefficients correspond to the complement of bottom-$(n_{\mathcal{I}} + n_{\mathcal{T}} - k)$ coefficients, which yields

$$M_{\mathbf{e},\min,n_{\mathcal{I}}+n_{\mathcal{T}}-k} = M_{n_{\mathcal{I}}+n_{\mathcal{T}}-k}^- = (N_{\mathcal{I}} \cup N_{\mathcal{T}}) \setminus M_k^+,$$

which finishes the proof of the deletion curve $\mathcal{C}_{\text{delete}} = \{(k, \nu(M_{\mathbf{e},\min,n_{\mathcal{I}}+n_{\mathcal{T}}-k}))\}_{k=1,\ldots,n_{\mathcal{I}}+n_{\mathcal{T}}}$.
$\square$

# B Implementation Details

## B.1 Faithful interaction explanations of LIP models (FIxLIP)

Because of how the model's forward function is implemented, without loss of generality, we explain a logit-scaled similarity output from Equation 1, i.e. $C \cdot f(x_\mathcal{I}, x_\mathcal{T})$, which depends on the particular model's learnt constant, e.g. $C \approx 100$ for CLIP, but $C \approx 117$ for SigLIP and $C \approx 112$ for SigLIP-2.

**Masking.** Following related work using input removal in attribution methods [17, 19, 27, 45], we apply simple baseline masking strategies, which were also proven successful based on our empirical results. For vision encoders, we use a **0** baseline after image normalization to propagate no signal forward. In the case of language encoders, we use a native attention masking mechanism, where we encode active tokens with 1 and the deleted ones with 0. It is important to leave the beginning/end sequence and padding tokens as is. Note that in theory, the maximum possible context length $n_\mathcal{T}$ is finite (e.g. 64, 77) and could be treated as constant across different inputs. However, we never explain these special tokens, treating parts of text inputs as the only features of interest. Alternative implementations of masking worth considering are: removing words from the text inputs, or tokens from tokenized inputs, imputing with the [UNK] token, or [MASK], e.g. in the case of the BiomedCLIP architecture that is based on BERT [76]. Acknowledging the potential influence of the masking approach, as well as position encodings, on the value of an empty input, we check that it remains about constant across different text input lengths in Figure 7.

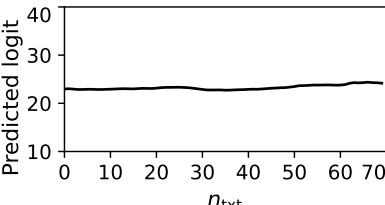

**Figure 7:** Value of an empty input is about constant across different text lengths.

**Budget split in the cross-modal estimator.** The model-agnostic estimator (Definition 6) relies on $m$ sampled masks for approximation; the number is often called the approximation budget. The cross-modal estimator (Definition 7) introduces budget split, where $m = m_\mathcal{T} \cdot m_\mathcal{I}$. For all our experiments, we set $m_\mathcal{T} := \min\left(2^{n_\mathcal{T}}, \max(4, \lceil \sqrt{m} \cdot n_\mathcal{T}/n_\mathcal{I} \rceil)\right)$ and $m_\mathcal{I} := \min\left(2^{n_\mathcal{I}}, \max(4, \lfloor \sqrt{m} \cdot n_\mathcal{I}/n_\mathcal{T} \rfloor)\right)$ as a reasonable default, allowing for a balanced exploration of both input spaces.

**Two-step filtering approach.** For the ViT-B/16 and ViT-L/16 model versions with a larger number of input image tokens (patches), we perform a cheap computation of the first-order attribution to prioritize interactions for approximation. We consider two strategies: picking a clique subset of the most interesting tokens, or focusing only on cross-modal interaction for explanation, which we specifically apply in the pointing game recognition (Definition 10, Appendix B.3). In the first strategy, given the desirable clique size $k = 72$, we take $k_\mathcal{T} := \max\left(5, \lceil k \cdot n_\mathcal{T}/(n_\mathcal{T} + n_\mathcal{I}) \rceil\right)$ text tokens with the highest absolute feature attribution scores, and $k_\mathcal{I} := k - k_\mathcal{T}$ image tokens accordingly. Similarly to the above considerations regarding the budget split between the two modalities, we found this to be empirically adequate. Future work can propose optimal strategies for both of these nuances.

**Greedy subset selection.** Even for the ViT-B/32 model version with fewer than 100 input tokens (nodes in a graph), finding cliques of the highest and smallest sums using brute force is computationally prohibitive. Still, we want to use these subsets denoted as $M_{\mathbf{e},\min,k}$ and $M_{\mathbf{e},\max,k}$ in evaluation with insertion/deletion curves (Definition 9) and explanation visualization. Thus, we implemented a simple greedy algorithm starting the search from each (or a subset) of the tokens (nodes). Its general goal is to add consecutive tokens to the subset based on minimizing/maximizing the subset's value. For models with a larger number of inputs, e.g. $n_\mathcal{I} + n_\mathcal{T} = 196 + 30$, the greedy strategy remains applicable; however, it may require a few minutes to complete when considering all subset sizes $k = 1, \ldots, n_\mathcal{I} + n_\mathcal{T}$.

## B.2 Area between the insertion/deletion curves (AID)

Figure 9 describes visually the process of computing insertion/deletion curves and how it can differ between the first-order attribution methods and second-order interaction explanations. Consider the following example shown in (**bottom, Ours, 73%**), where the faces of a smiling man and child are masked with the masked caption saying "a man — a child smiling at a — in a —". Here, the model predicts the input to be dissimilar (below-average similarity), which means it enters negative values in our normalized case. Contrarily, in (**bottom, Baseline**), the unmasked `restaurant` text token keeps the model's similarity near the average level. Another interesting phenomenon happens for the insertion curve in Figure 9 (**top**). Faithfully masking the redundant information with **our** method causes the model's similarity to increase above the original prediction. Contrarily, in (**top, Baseline**), the baseline gradient-based method is unable to faithfully recover such redundant tokens.

## B.3 Pointing game recognition (PGR)

Figure 8 illustrates an example of the pointing game evaluation for interaction explanations like FIxLIP, giving additional context as to why first-order attribution methods fail to pass this desirable sanity check. A saliency map can only highlight important parts of image/text inputs, but whenever more objects/concepts appear in the input, they are unable to disentangle the basic relationship between the two modalities. PGR metric measures the ratio of absolute values of "correctly" identified cross-modal interaction terms to total interaction terms. For example, in Figure 8 (**2nd row, 2nd column**), we sum the positive interactions between "banana"/"cat" image tokens and the `banana`/`cat` text tokens, as well as the absolute negative interactions between "tractor"/"ball" image tokens and the `banana`/`cat` text tokens. Then, we divide this sum by the total absolute sum of all interactions, which gives a normalized PGR score. In Appendix C.2, we further describe the specific combinations of image and object/class labels used to create our exemplary benchmark in this paper, although the overall methodology should be treated in a generic manner.

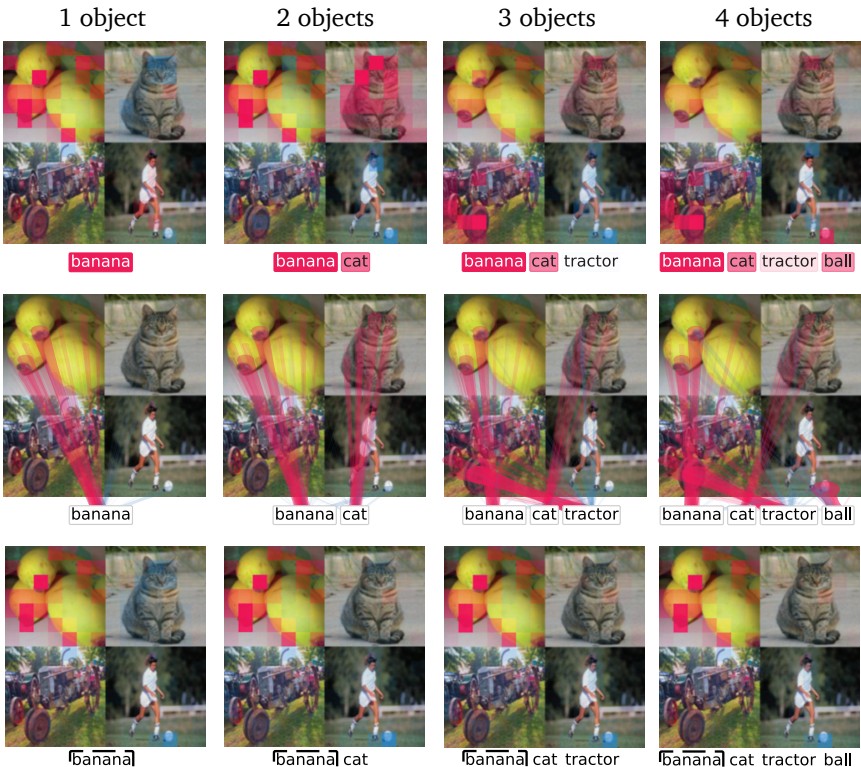

**Figure 8:** Visualization of a first-order attribution (**top row**), e.g. Shapley values, compared to cross-modal interactions in FIxLIP (**middle row**) for an exemplary pointing game. FIxLIP can be conditioned on any token, e.g. the first text input token (**bottom row**) for a simpler visualization. Consecutive columns here correspond to consecutive columns in Tables 1, 2 & 4.

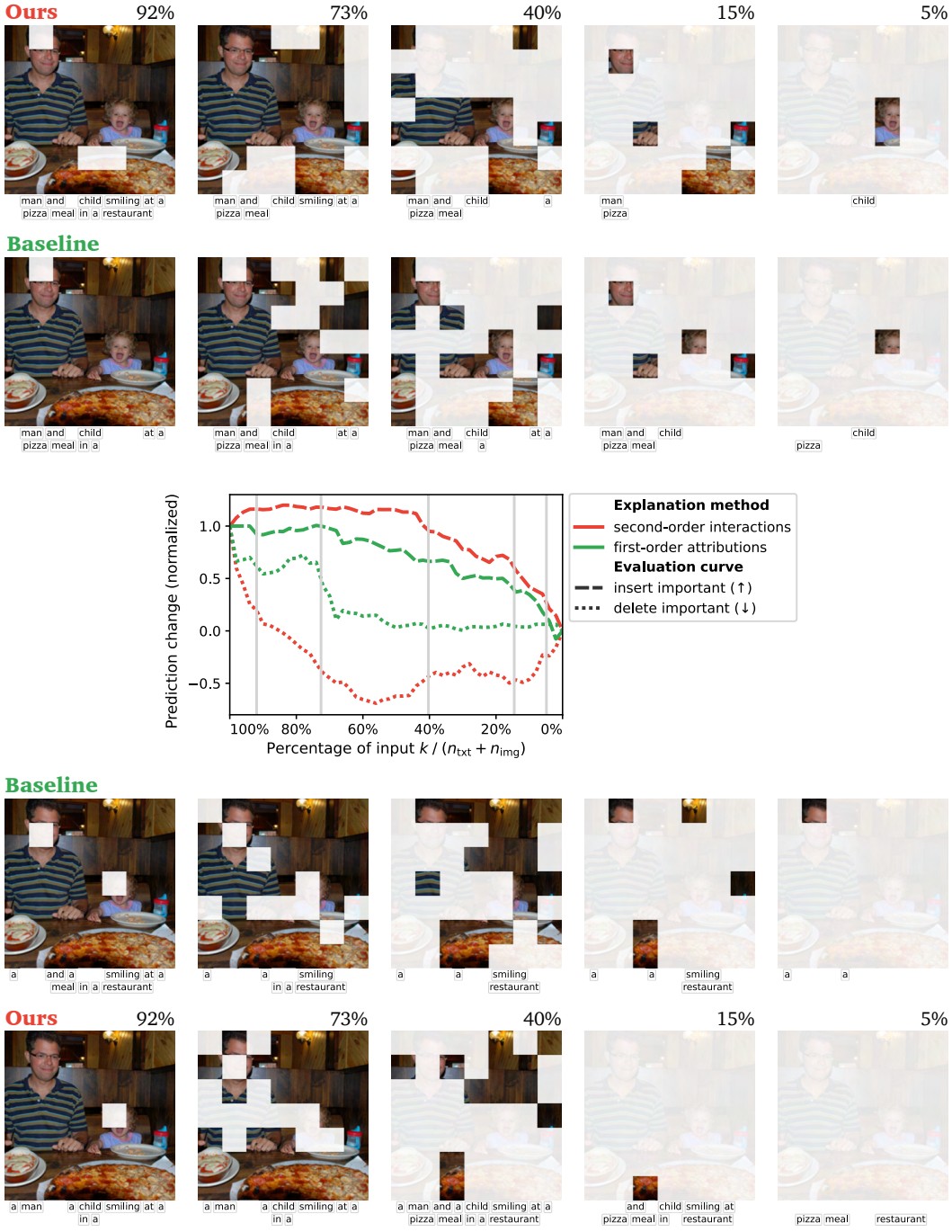

**Figure 9:** Exemplary insertion/deletion curves for two explanations **e** computed with FIXLIP and another first-order method. We show five subset visualizations per curve per method. In the case of the baseline method, each consecutive subset needs to include the preceding one, which is not the case for FIXLIP, capturing the complex cross-modal and intra-modal interactions. For example, a `child` with the corresponding face patches (**top right corner**) gets overvalued by the `man–pizza` tokens appearing jointly with the corresponding image part (`child` disappears). Similarly, in the deletion curve, the subsets of text and image tokens interchange with each other at around 50% (**bottom row**).

# C Experimental Setup

## C.1 Models

We use the following openly available models from Hugging Face with default hyperparameters:

- CLIP ViT-B/32 [57]: `openai/clip-vit-base-patch32` (MIT License)
- CLIP ViT-B/16 [57]: `openai/clip-vit-base-patch16` (MIT License)
- SigLIP ViT-B/16 [75]: `google/siglip-base-patch16-224` (Apache License 2.0)
- SigLIP ViT-L/16 [75]: `google/siglip-large-patch16-256` (Apache License 2.0)
- SigLIP-2 ViT-B/32 [66]: `google/siglip2-base-patch32-256` (Apache License 2.0)
- SigLIP-2 ViT-B/16 [66]: `google/siglip2-base-patch16-224` (Apache License 2.0)
- SigLIP-2 ViT-L/16 [66]: `google/siglip2-large-patch16-256` (Apache License 2.0)

We additionally rely on the CLIP ViT-B/32 and ViT-B/16 model versions from the `clip` Python library [57, MIT License], not to modify the official implementation of the baselines (Appendix C.3). We set the batch size to 64 for the base models and to 32 for the large models, performing computation on A100 GPUs with 40GB VRAM.

## C.2 Datasets

We use the following openly available datasets from Hugging Face:

- MS COCO test set [42]: `clip-benchmark/wds_mscoco_captions` (CC BY 4.0)
- ImageNet-1k validation set [15]: `ILSVRC/imagenet-1k` (ImageNet Agreement)

Experiments with the CLIP models are performed using 1000 image–text pairs from the MS COCO test set, for which each of the models predicted the highest similarity scores. Experiments with the SigLIP-2 models are performed using 100 image–text pairs to save computational resources, since we are not using it to compare with baseline methods.

Regarding ImageNet-1k, we use all 50 images from each of the following 10 class labels for constructing the pointing game evaluation: `goldfish` (1), `husky` (248), `cat` (282), `plane` (404), `church` (497), `ipod` (605), `ball` (805), `tractor` (866), `banana` (954), `pizza` (963). We combine these images with labels into 10 varied games as follows: `goldfish-husky-pizza-tractor`, `cat-goldfish-plane-pizza`, `banana-cat-tractor-ball`, `husky-banana-plane-church`, `pizza-ipod-goldfish-banana`, `ipod-cat-husky-plane`, `tractor-ball-banana-ipod`, `plane-church-ball-goldfish`, `church-pizza-ipod-cat`, `ball-husky-banana-tractor`. In each case, we distinguish four scenarios by gradually adding each consecutive token from left to right, resulting in 40 scenarios, each with a total of 50 images, which constitutes a reasonable case.

## C.3 Baselines

We use the following publicly available source code:

- GAME [12]: https://github.com/hila-chefer/Transformer-MM-Explainability (MIT License)
- Grad-ECLIP [77]: https://github.com/Cyang-Zhao/Grad-Eclip (License unknown)
- exCLIP [48]: Supplementary material at https://openreview.net/forum?id=plkrRJt98c (License unknown)

We were unable to run exCLIP on the CLIP (ViT-B/16) model version for inputs with more than 24 text tokens, which are common in MS COCO, running into out-of-memory errors. Furthermore, we refrain from attempting to run these implementations on SigLIP models, which were not part of the original experiments. For context, we did not find an openly available implementation of InteractionLIME [32], while it is not clear how to adapt InteractionCAM [61] (proposed for image–image encoders) to the text/language domain.

## C.4 Metrics

For $p$-faithfulness correlation (Definition 8), we set $m = 1000$ for a reasonable sample size and analyze $p = \{0.3, 0.5, 0.7\}$, as well as Shapley sampling weights to give a broader context.

Note that the insertion/deletion curves (Definition 9) have different lengths for each input (explanation), depending on the number of tokens (players). We thus interpolate each curve over a fixed range of 51 points between 0–100% input and then aggregate them to obtain the final visualization.

Pointing game recognition has no additional hyperparameters beyond the pre-defined dataset described in Appendix C.2. For the example presented in Table 2, we omit the (sub)games with either `goldfish` or `ipod` for SigLIP models, because their corresponding text tokenizer divides these class labels into multiple tokens.

## C.5 Compute resources

Experiments described in Section 5 and Appendix D were computed on a cluster consisting of $4\times$ AMD Rome 7742 CPUs (256 cores), 4TB of RAM, and $16\times$ A100 GPUs for about 15 days combined. We envision that preliminary and failed experiments have required the same amount of compute resources.

## D    Additional Experimental Results

We report the insertion/deletion results for CLIP (ViT-B/16) in Figure 10 and for SigLIP-2 (ViT-B/32) in Figure 11, where we also analyze the gradual relationship between $p$-masking in FIxLIP and the model's prediction change more gradually. Table 4 reports pointing game results for CLIP (ViT-B/16), where interestingly FIxLIP with $p = 0.5$ outperforms FIxLIP (Shapley interactions).

Figure 12 shows further ablations based on the different distributions in the $p$-faithfulness correlation metric. Interestingly, FIxLIP with $p = 0.7$ evaluated on masks from $p = 0.3$ performs much better than FIxLIP with $p = 0.3$ evaluated on masks from $p = 0.7$, which further confirms our motivation to increase $p$ in machine learning applications where out-of-distribution sampling is not desirable.

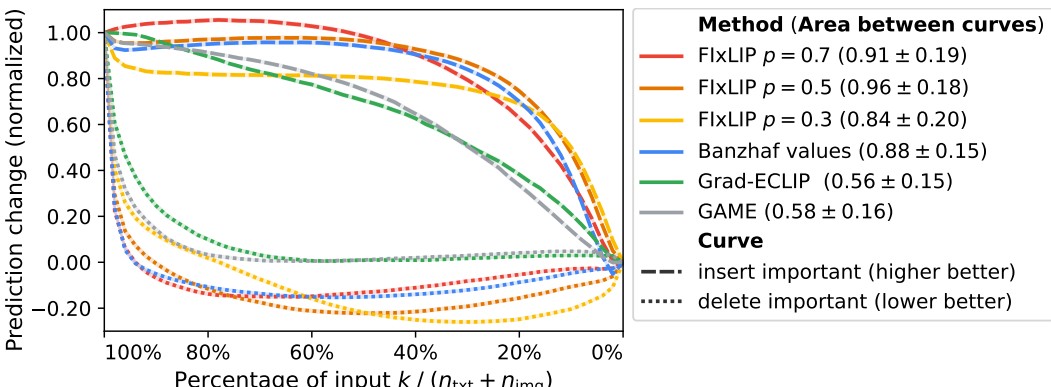

**Figure 10: Insertion/deletion curves for CLIP (ViT-B/16) on MS COCO.** Extended Figure 3. AID score (higher is better) for FIxLIP against alternative explanation methods, where a random baseline scores 0. The y-axis is normalized between the model's prediction on the original input (100%) and the fully removed one (0%), where negative values denote that the model is predicting the inputs are unsimilar. Gradient-based methods such as Grad-ECLIP fail to recover nonlinear rankings of important tokens, whereas FIxLIP faithfully recovers the optimal subset explanation. exCLIP does not appear in the comparison as its implementation returns an OOM error when querying the larger model ($14 \times 14$ image tokens) with texts longer than 24 tokens, which are common in MS COCO.

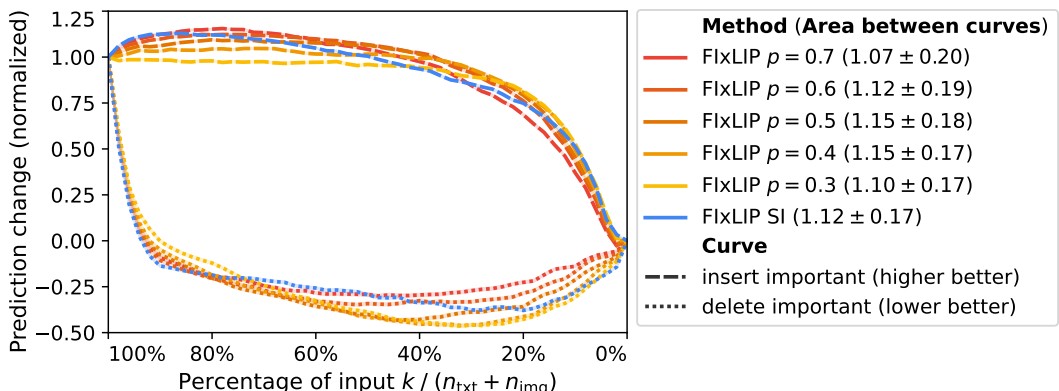

**Figure 11: Insertion/deletion curves for SigLIP-2 (ViT-B/32) on MS COCO.** Extended Figure 3. AID score (higher is better) for a model-agnostic estimator of FIxLIP with different masking weights. The y-axis is normalized between the model's prediction on the original input (100%) and the fully removed one (0%), where negative values denote that the model is predicting the inputs are unsimilar. We observe a gradual relationship between $p$ and the steepness of the curves at a particular fraction of input deleted. We omit comparison with other baselines in this case as they were originally implemented for CLIP and not SigLIP.

**Table 4: Pointing game recognition for CLIP (ViT-B/16) on ImageNet-1k.** Extended Table 1. PGR score for FIxLIP against alternative explanation methods, where a random baseline scores 0.25. First-order methods, such as Grad-ECLIP and Shapley values, fail to distinguish between multiple objects simultaneously, while second-order methods faithfully recover the appropriate explanation (up to the irreducible non-optimality of the pointing game).

| Explanation Method | Recognition ($\uparrow$) | | | |
| --- | --- | --- | --- | --- |
| | 1 object | 2 objects | 3 objects | 4 objects |
| GAME [12] | $.60_{\pm.11}$ | $.41_{\pm.04}$ | $.33_{\pm.02}$ | $.27_{\pm.01}$ |
| Grad-ECLIP [77] | $.72_{\pm.13}$ | $.44_{\pm.04}$ | $.33_{\pm.01}$ | $.26_{\pm.01}$ |
| Shapley values | $.66_{\pm.07}$ | $.57_{\pm.05}$ | $.50_{\pm.04}$ | $.42_{\pm.03}$ |
| Banzhaf values | $.70_{\pm.08}$ | $.58_{\pm.05}$ | $.50_{\pm.05}$ | $.41_{\pm.04}$ |
| exCLIP [48] | $.73_{\pm.15}$ | $.82_{\pm.07}$ | $.85_{\pm.05}$ | $.87_{\pm.05}$ |
| FIxLIP (Shapley interactions) | $.76_{\pm.08}$ | $.78_{\pm.06}$ | $.79_{\pm.05}$ | $.79_{\pm.05}$ |
| FIxLIP (w. Banzhaf interactions $p = 0.3$) | $.76_{\pm.10}$ | $.75_{\pm.08}$ | $.77_{\pm.07}$ | $.77_{\pm.06}$ |
| FIxLIP (w. Banzhaf interactions $p = 0.5$) | $.81_{\pm.08}$ | $.80_{\pm.06}$ | $.81_{\pm.05}$ | $.81_{\pm.05}$ |
| FIxLIP (w. Banzhaf interactions $p = 0.7$) | $.75_{\pm.09}$ | $.75_{\pm.07}$ | $.77_{\pm.06}$ | $.78_{\pm.05}$ |

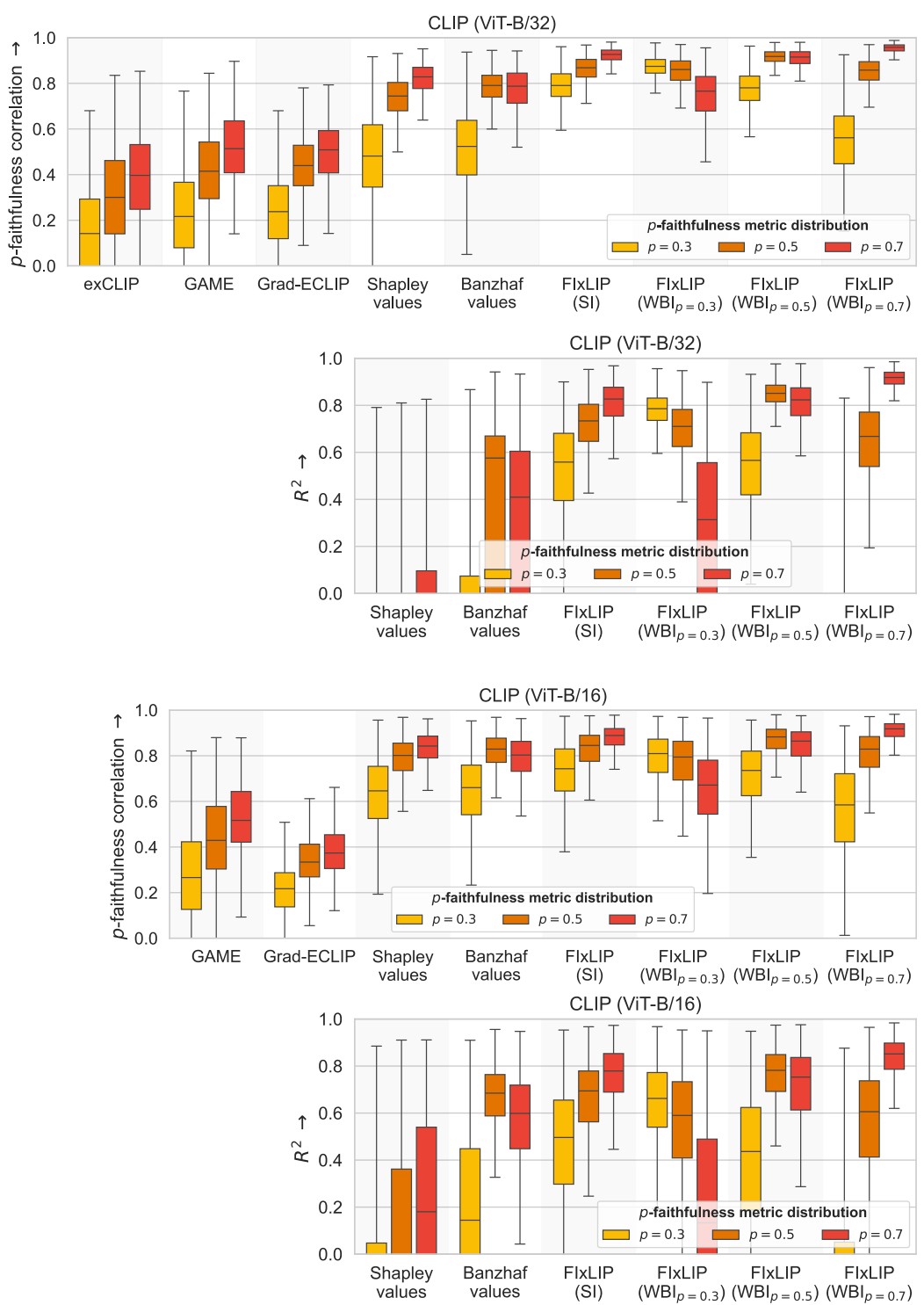

**Figure 12: Faithfulness evaluation for CLIP (ViT-B/32 and ViT-B/16) on MS COCO**. Extended Figure 4. We evaluate the faithfulness of different FIxLIP variants and baselines in terms of $p$-faithfulness correlation and the $R^2$ coefficient (where it is possible for game-theoretic approaches). For 1 000 inputs, we sample 1 000 evaluation masks according to different $p$-faithfulness distributions: $p = 0.3$, $p = 0.5$, $p = 0.7$. For each $p$-faithfulness distribution, the corresponding FIxLIP (w. Banzhaf interactions) outperforms the baselines.

# E   Visual Explanatons

## E.1   Guidance on interpreting visualizations of interaction explanations

In all of our visualizations, explanations are highlighted using a red–blue color scheme, where red indicates a positive contribution to the model's output and blue a negative one. The transparency of the color reflects the strength of the contribution—more opaque elements correspond to higher absolute explanation values. Pairwise interactions are represented as edges, main effects (or first-order attributions) are displayed as colored boxes around words or overlayed as heatmaps on top of images. In addition to the color and transparency, edges also vary in thickness based on the interaction strength. In general, interactions may link elements within the same modality, e.g. text–text interactions or image–image interactions, or across modalities, e.g. text–image interactions.

Figure 13 illustrates how to interpret visualizations in a multimodal setting involving image and text inputs. Provided an input image and text pair (Figure 13**A**), we compute FIxLIP and retrieve an explanation $\mathbf{e}$ containing interactions and main effects. The FIxLIP output (Figure 13**B**) can be visualized by plotting the image and text together, and overlaying the main effects and interactions. This allows for a quick screening of interesting parts in the input as judged by an explanation. In this particular example, the `elephant` text token is positively linked to all image tokens containing the elephant. The two text tokens `white` and `bird` are linked to the one image token containing the bird. Here, `elephant` is red because removing it would decrease the predicted similarity score, while `bird` is blue because removing it might even increase the predicted similarity score (depending on cross-modal interactions). Based on the FIXLIP, more detailed visualizations can be created.

**Conditioning on tokens.** To investigate the role of a specific token $i \in N_\mathcal{T} \cup N_\mathcal{I}$, we can condition the interaction space on $i$ and retrieve all interactions of the form $\mathbf{e}_{\{i,j\}}$ for $j \in N \setminus \{i\}$. This conditional view highlights how token $i$ interacts with all other tokens and is especially useful for analyzing its contextual relevance across modalities. Then, we project the interactions onto the first-order and visualize them with a heatmap (Figure 13**C**). Conditioning on `elephant` shows the greatest positive interactions with the body in the image tokens and the word `small`. Conditioning on the image token containing the bird mostly interacts with `white` and `bird`.

**Finding interesting subsets.** FIxLIP explanations allow searching for interesting subsets $M$ like $M_{\mathbf{e},\max,k}$ or $M_{\mathbf{e},\min,k}$ for varying sizes of $k$. We can search the space of interactions and visualize only the explanations for tokens in these subsets, or overlay the masks on top of the input. In Figure 13**D**, we illustrate $M_{\mathbf{e},\max,7}$ containing tokens related to `elephant`. Starting the search from the `bird` token retrieves the $k = 4$ subset with information about the bird. While the elephant subset increases the similarity, the bird subset could slightly decrease the similarity, presumably because the model would find the isolated image–text similarity of the elephant clearer.

**Visualizing heatmaps.** Based on Theorem 1, second-order explanations provided by FIxLIP can be converted into first-order explanations. Hence, FIxLIP explanations can also be visualized with heatmaps (Figure 13**E**). Similar to baselines such as Grad-ECLIP or Shapley values, we overlay the converted first-order FIxLIP explanations onto the input. The scores are scaled *jointly* across both modalities since we retrieve the explanation scores across both modalities. An alternative could be to scale scores *independently* with different maximum values (one per modality). Joint scaling allows for relative comparisons, which is not the case for independent scaling, where two tokens are visualized with full color intensity. In Figure 13**E** (**left**), the independently scaled heatmap shows that the vision token containing the bird and the `elephant` text token have the highest scores in their respective modality. Jointly scaling the heatmap reveals that the `elephant` text token is of a higher score than the vision token containing the bird.

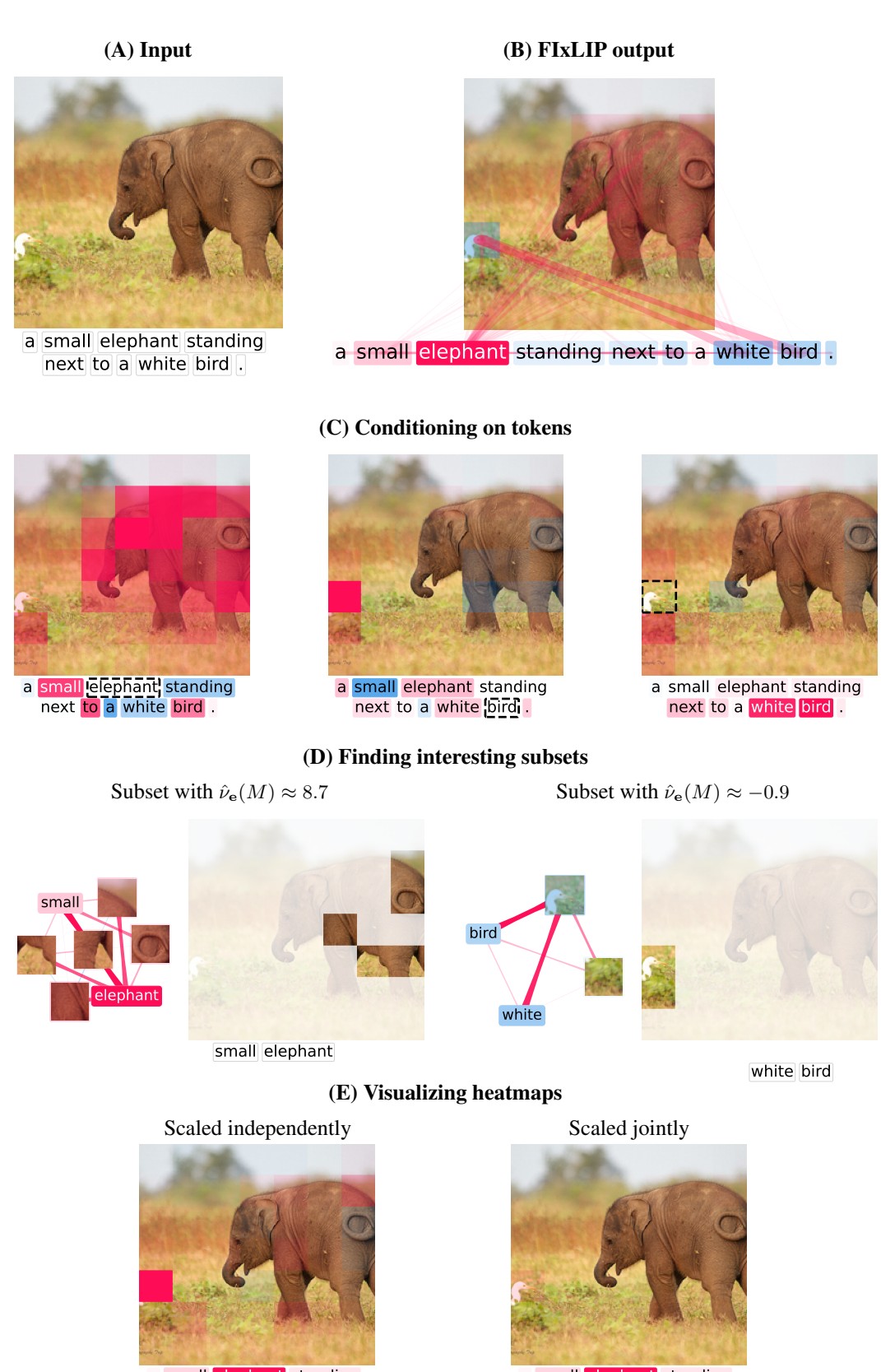

**Figure 13: Showcase of explanation visualization.** A comprehensive description is in Appendix E.1.

## E.2 Additional examples

Figure 14 demonstrates the utility of conditioning on tokens. Figure 15 shows intriguing examples of interactions in inputs with text written in an image, which is related to the typographic capabilities of large vision–language models [10, 23, 63].

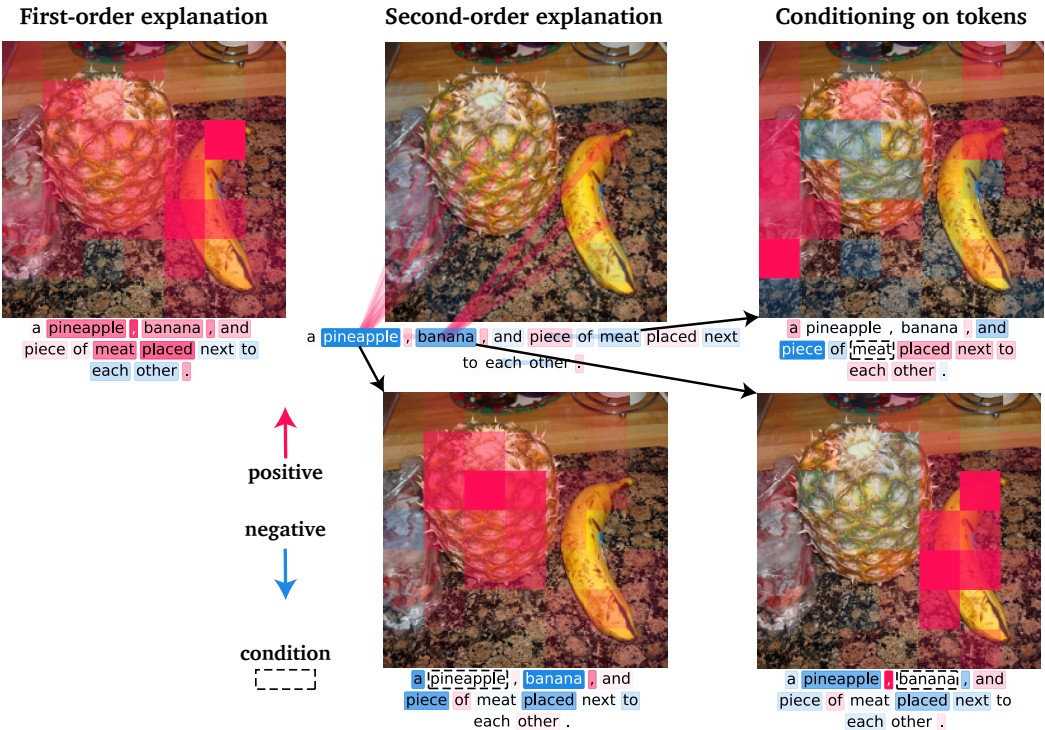

**Figure 14: FIxLIP provides a utility of conditional heatmap generation.** Comparison between a first-order attribution explanation and a second-order interaction explanation, which is able to recognize complex nuances of the presented scene.

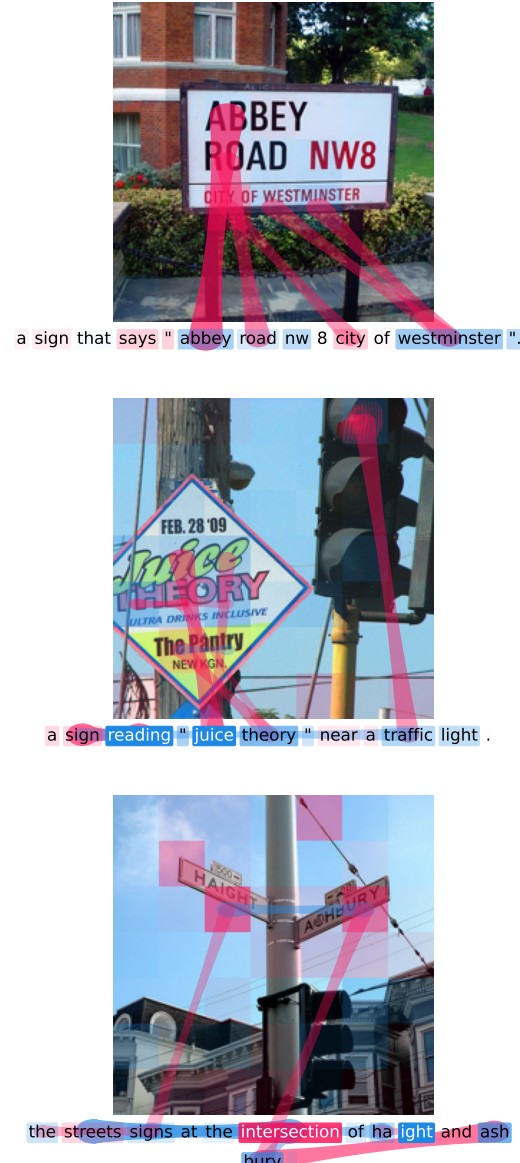

**Figure 15: Intriguing examples.** Top interactions in inputs with text written in an image.

### E.3 Comparison between CLIP (ViT-B/32) and SigLIP-2 (ViT-B/32)

See Figure 16. Note that these models have different text tokenizers and input image resolutions, i.e. $224 \times 224$ vs. $256 \times 256$, which is why the images differ slightly.

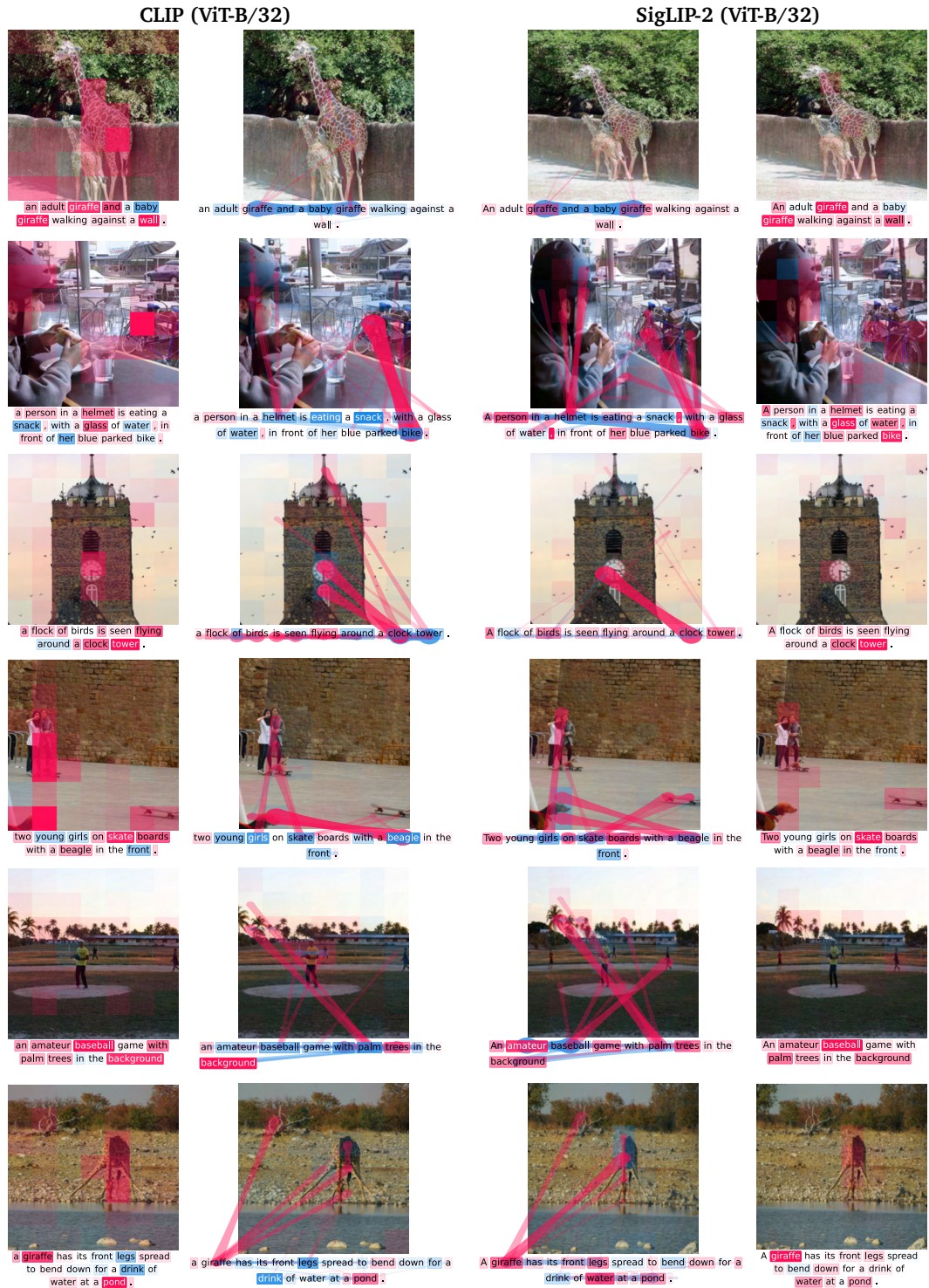

**Figure 16:** Visual comparison between FIxLIP of CLIP (**left**) and SigLIP-2 (**right**).

## E.4 Comparison between CLIP (ViT-B/32) and CLIP (ViT-B/16)

Visual examples between CLIP (ViT-B/32) and CLIP (ViT-B/16) are in Figure 17. Notably, in the example including the clock (Fig. 17, **row 3**), conditioning on the vision token containing the letters T and part of O reveals a strong interaction with another vision token of this word in the "sign" *and* the text token town. This may suggest the presence of a *third-order* interaction between these tokens.

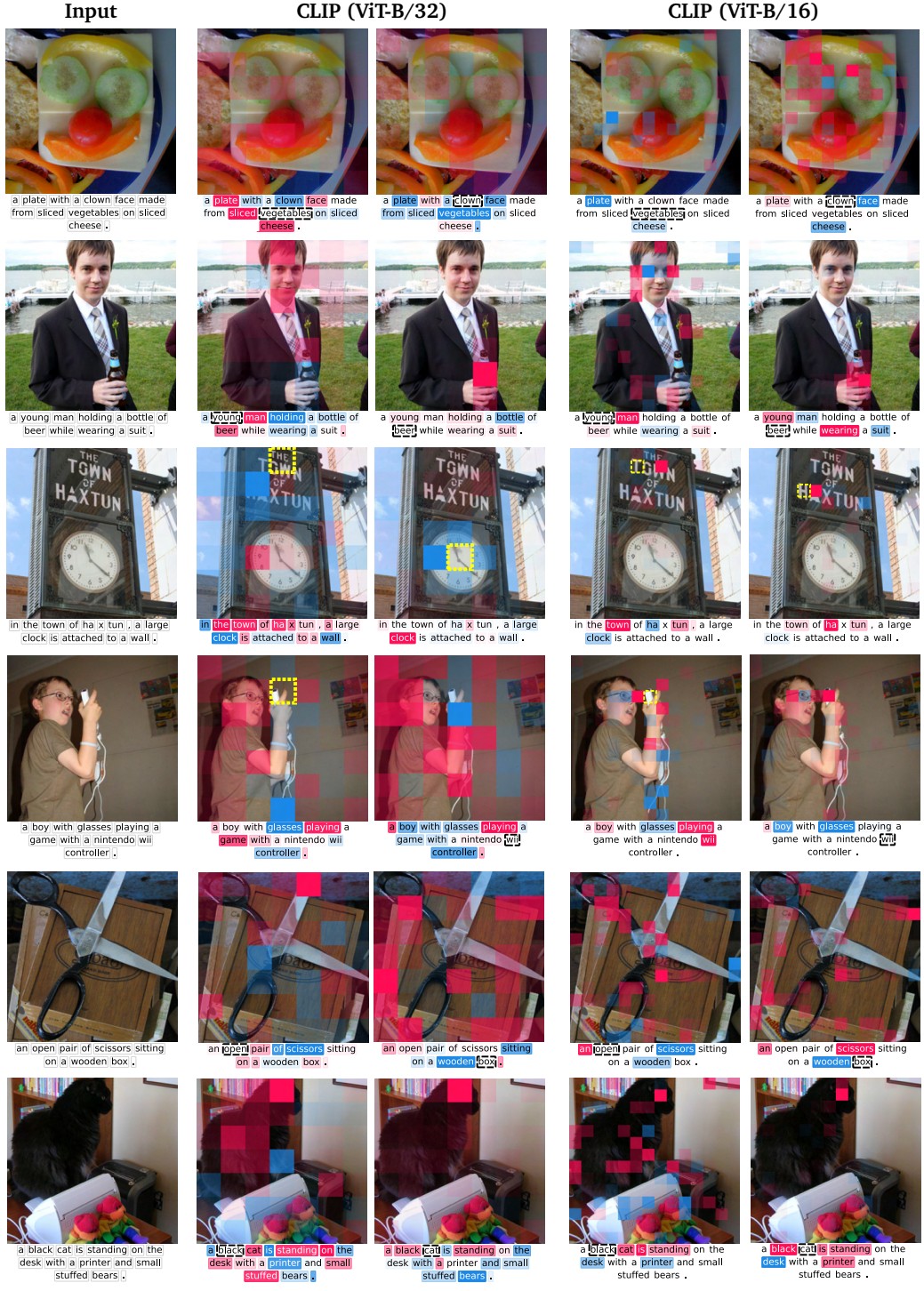

**Figure 17:** Visual comparison between FIxLIP of CLIP ViT-B/32 (**left**) and ViT-B/16 (**right**). Each heatmap is conditioned on the selected input token (in black), also for vision tokens (in yellow).

