# OpenReview forum: "Explaining Similarity in Vision-Language Encoders with Weighted Banzhaf Interactions"
_NeurIPS.cc/2025/Conference — NeurIPS 2025 poster_

### Official Review · Reviewer_WGoS · 2025-06-25

**Clarity:** 1
**Significance:** 3
**Originality:** 3
**Rating:** 4
**Confidence:** 3

**Summary:**

This paper points out that the current explanation methods with saliency maps are limited by the unimodal view of the model’s output. The vision-language encoders should be explained by considering the cross-modal interactions between image-text pairs. Therefore, the paper proposes an explanation method to measure the importance of each token in the image and text encoders via weighted faithful Banzhaf interactions.

**Questions:**

- It is confusing that the explanation for the two-modality interaction is called second-order. If there is a definition, it should mention the reference.
- It said that this is the first work using weighted Banzhaf interactions to overcome the out-of-distribution problem apparent in removal-based explainability in line 46. How to understand the out-of-distribution? It’s better to have a clear demonstration in the introduction.
- I know the attribution-based methods like Grad-ECLIP and GAME are totally different from the methods with game theory. Then, what’s the main difference and the advantages of using Banzhaf compared with Shapley to estimate the token importance? It will be clearer to have a figure to show the comparison of the two kinds of algorithms.
- For the explanation graph in Figure 1, why do the tokens of object parts have a negative contribution to the similarity, such as the two dog patches and the three hydrant patches? More demonstrations about the results in Figure 1 should be added. It’s hard to understand.

About experiments:
- In Figure 2, negative y-axis values can be obtained with the Sharpley and FIXLIP methods, while the minimum values for the attention heat map methods (Grad-ECLIP, GAME, and XCLIP) are zero. This will greatly increase the performance gap. The reason given in the paper is “negative values denote that the model is predicting the inputs are unsimilar.” What does that mean? Another image is also used in the deletion/insertion experiments? I'm still confused about why the similarity drop can be negative.
- What is the concrete process in the PGR metric and sanity check in Section 5.2? I know that the point game is that a hit shows when the maximum value on the heat map is located on the correct object. So, what’s the operation when multiple objects are involved? And what is the sanity check with a pointing game?

I may increase my rating after my concerns are solved.

**Ethical Concerns:**

["NO or VERY MINOR ethics concerns only"]

**Final Justification:**

Thank the authors for their thorough replies. My concerns are basically solved, so I would like to raise my rating to 4.

**Limitations:**

Yes.

**Quality:**

2

**Strengths And Weaknesses:**

Strengths:
- Very comprehensive algorithm demonstration.
- It's the first work to use game theory to generate explanations for cross-modal interaction in the vision-language models.

Weaknesses:
- The clarity of the paper needs to be improved. Concretely, please see the questions.

---

> ### Author Rebuttal · Authors · 2025-07-29
>
> **We gratefully thank the reviewer for a thorough and insightful review of our work.** Below, we provide point-by-point responses to each of the questions.
>
> > **Q1:** It is confusing that the explanation for the two-modality interaction is called second-order. If there is a definition, it should mention the reference.
>
> Following the closely related literature [35, 46], we use *second-order* to denote interaction scores between *two* tokens (features). Please note that a *second-order* interaction is a subset of a more general term called *higher-order* or *any-order* interaction scores between feature subsets of *higher* or *any* size, following the related work [16, 28, 44, 58].
>
> > **Q2:** It said that this is the first work using weighted Banzhaf interactions to overcome the out-of-distribution problem apparent in removal-based explainability in line 46. How to understand the out-of-distribution? It’s better to have a clear demonstration in the introduction.
>
> Thank you for this valuable feedback, which we will use to improve the introduction. We understand out-of-distribution data as inputs with many masked tokens (**line 118**), where images become unrecognizable and text captions are ambiguous. We show an example of such an out-of-distribution input in **Figure 1**, where for $p=0.3$ most of the image–text pair is masked and unrecognizable to the model. Our method addresses the emerging requirement for controllable sparsity in mask sampling [5, 21, 23, 71] (**line 39**).
>
> > **Q3:** I know the attribution-based methods like Grad-ECLIP and GAME are totally different from the methods with game theory. Then, what’s the main difference and the advantages of using Banzhaf compared with Shapley to estimate the token importance? It will be clearer to have a figure to show the comparison of the two kinds of algorithms.
>
> We highlight the two main differences between Banzhaf and Shapley in **Remark 2**. FIxLIP with $p$-weighted (Banzhaf) masking allows using the cross-modal estimator (**Definition 7**), which is far more efficient with respect to budget, i.e. obtains $m^2$ model inferences from $m$ input samples, as depicted with $m \rightarrow m^2$ in **Figure 1**. Therefore, its *main advantage is computational efficiency*. Furthermore, it allows for prioritizing masks that are expected to better reflect in-distribution inputs. Yet, both Banzhaf and Shapley provide faithful explanations. We implement FIxLIP with Shapley using the baseline model-agnostic estimator (**Definition 6**), which obtains only $m$ model inferences from $m$ input samples instead. Note that we consider both approaches to be a contribution of this paper and compare them in experiments as appropriate.
>
> > **Q4:** For the explanation graph in Figure 1, why do the tokens of object parts have a negative contribution to the similarity, such as the two dog patches and the three hydrant patches? More demonstrations about the results in Figure 1 should be added. It’s hard to understand.
>
> Critically questioning the model’s reasoning, as demonstrated by the Reviewer’s curiosity in this point, is precisely the motivation for this paper. *The behaviour of vision–language encoders like SigLIP-2 can be highly complex*. Thus, although our contribution is to improve their understanding through interaction explanations that are measurably faithful to the similarity prediction, we cannot expect the model to be aligned with human intuition. In the case from **Figure 1**, we see that intra-modal interactions between image tokens are relatively unimportant as compared to the intra-modal interactions between text tokens and cross-modal ones. We hypothesize that the dog and hydrant image patches carry a lot of redundant information. Thus, interactions between these tokens can be slightly negative as compared to the strong ‘hydr’–‘ant’ text interaction and cross-modal interactions. Note that the graph-like visualization of explanations is *only one of the three approaches* we showcase in **Figure 5**. We also provide a visual comparison between FIxLIP and baseline methods using the dog/hydrant example in **Appendix E.2, Figure 13**. Finally, to aid in the understanding of the introduced visualizations, we provide a comprehensive guide on interpreting them in **Appendix E.1**.
>
> > **Q5:** In Figure 2, negative y-axis values can be obtained with the Sharpley and FIXLIP methods, while the minimum values for the attention heat map methods (Grad-ECLIP, GAME, and XCLIP) are zero. This will greatly increase the performance gap. The reason given in the paper is “negative values denote that the model is predicting the inputs are unsimilar.” What does that mean? Another image is also used in the deletion/insertion experiments? I'm still confused about why the similarity drop can be negative.
>
> This is a great question. In theory, all of the methods (Grad-ECLIP, GAME, XCLIP, FIxLIP) can obtain negative normalized values on the Y-axis. *Negative values denote that the model is predicting the image–text pair to be unsimilar*. It means that the prediction on a masked input is smaller than the prediction on the empty input (see **Appendix B.1, Figure 6**). We give a supplementary explanation of the insertion/deletion process on a single image in **Appendix B.2, Figure 7**. Consider the following example shown in **Figure 7 (bottom, Ours, 63%)**, where the faces of a smiling man and child are masked with the masked caption saying ‘a man — a child smiling at a — in a —’. Here, the model predicts the input to be dissimilar (below-average similarity), which means going into negative values in our normalized case. Contrarily, in **Figure 7 (bottom, Baseline)**, the unmasked ‘restaurant’ text token keeps the model’s similarity near the average level. Another interesting phenomenon happens for the insertion curve in **Figure 7 (top)**. Faithfully masking the *redundant* information with **our** method causes the model’s similarity to *increase* above the original prediction. Contrarily, in **Figure 7 (top, Baseline)**, the baseline gradient-based method is unable to faithfully recover such redundant tokens. Finally, note that **Figures 2, 9 & 10** show an average over a set of such examples given in **Figure 7**.
>
> > **Q6:** What is the concrete process in the PGR metric and sanity check in Section 5.2? I know that the point game is that a hit shows when the maximum value on the heat map is located on the correct object. So, what’s the operation when multiple objects are involved? And what is the sanity check with a pointing game?
>
> Kindly note that we provide a comprehensive description of this process in **Appendix B.3, Figure 8**. When multiple objects are involved, the pointing game recognition (PGR) metric measures the ratio of absolute values of “correctly” identified cross-modal interaction terms to total interaction terms. For example, in **Figure 8 (2nd row, 2nd column)**, we sum the positive interactions between ‘banana’/‘cat’ image tokens and the ‘banana’/‘cat’ text tokens, as well as the absolute negative interactions between ‘tractor’/‘ball’ image tokens and the ‘banana’/‘cat’ text tokens. Then, we divide this sum by the total absolute sum of all interactions, which gives a normalized PGR score. By performing the *sanity check*, we check that interaction explanations are essential for explaining vision–language encoders as compared to baseline attribution approaches. Scoring high PGR values across 1, 2, 3, and 4 objects in **Table 1** denotes that an explanation method can distinguish between multiple objects at once, thus passing the sanity check.
>
> > I may increase my rating after my concerns are solved.
>
> **Thank you for taking this into consideration.** We would greatly appreciate an increase in rating if you feel we adequately addressed your questions. Else, we would appreciate an opportunity to address any remaining concerns.

---

### Official Review · Reviewer_Raws · 2025-06-30

**Clarity:** 3
**Significance:** 3
**Originality:** 3
**Rating:** 4
**Confidence:** 3

**Summary:**

This paper proposes FIXLIP for interpreting similarity scores in vision–language encoders through second-order interaction attributions grounded in weighted Banzhaf interaction indices. The authors address the limitations of existing first-order saliency methods by modeling cross- and intra-modal token interactions, thus capturing richer and more faithful explanation structures. Experiments show that FIXLIP produces more faithful and informative explanations than prior state-of-the-art attribution methods.

**Questions:**

See weakness.

**Ethical Concerns:**

["NO or VERY MINOR ethics concerns only"]

**Final Justification:**

This is a borderline paper to me while I lean to accept this paper. My biggest concerns are 1) the proposed method seems to be a  simple extension and 2) the budget might be a trouble in explanining models. An okay/good enough pape to me personally.

**Limitations:**

Yes.

**Paper Formatting Concerns:**

No.

**Quality:**

3

**Strengths And Weaknesses:**

Strengths:
1. Authors extend explainability to second-order interactions using weighted Banzhaf indices, offering more expressive and faithful attribution than first-order approaches.
2. Strong theoretical formulation.
3. The experimental results show the superiority of the proposed second-order estimator,

Weakness:
1. What is WLS in Figure 1?
2. Simply masking the patches and subtokens could be a bad choice as the final representations of patches and subtokens are entangled with each other. Also I assume masking subtokens means remove them from the sentence. But what does masking mean for patches? Do authors mask them as a whole black or white patch?
3. Although second-order interactions are insightful, the method does not scale to third-order or beyond, which might limit expressivity in complex VLMs. A simple case could be just have the third-order established by edge transformation.
4. It would be interesting to see how FlixLIP compared to previous works given the same budget.
5. How does the subset selection affect performance (Sec. 3.3)?
6. What are the computational overhead for different p settings in Table 1? Also what is the budget for other methods.

---

> ### Author Rebuttal · Authors · 2025-07-29
>
> **We sincerely thank the reviewer for their appreciation of our work and the constructive feedback.**
>
> > **Strengths:** Authors extend explainability to second-order interactions using weighted Banzhaf indices, offering more expressive and faithful attribution than first-order approaches. Strong theoretical formulation. The experimental results show the superiority of the proposed second-order estimator,
>
> Thank you for highlighting **our theoretical contribution**!
>
> > **W1:** Simply masking the patches and subtokens could be a bad choice as the final representations of patches and subtokens are entangled with each other. Also I assume masking subtokens means remove them from the sentence. But what does masking mean for patches? Do authors mask them as a whole black or white patch?
>
> We agree that optimizing the masking technique for transformers is an open research question. We view this as an implementation detail of our proposed faithful interaction explanations, where one can implement it with any masking technique proposed in the literature. Kindly note that we mention implementing masking text tokens with *attention masking* (**lines 103–104**), and *not removing them from the sentence*, which indeed could be a bad choice. Regarding image patches, we mask them with a $0$ baseline value (after image normalization) to propagate no signal forward. These and further implementational details can be found in **Appendix B.1**.
>
> > **W2:** Although second-order interactions are insightful, the method does not scale to third-order or beyond, which might limit expressivity in complex VLMs. A simple case could be just have the third-order established by edge transformation.
>
> While we (truly) think that this is indeed an interesting direction, we think that it does not invalidate our method and positive results. Our work takes a first big step in offering a unique perspective on interpreting image–text similarity predictions through second-order interactions. While the weighted Banzhaf interactions can be naturally extended to third-order explanations, this entails computational and perceptual challenges. Note that we state *efficiently approximating higher-order interactions* as an important future work direction in **line 323**. Take for example the image–text pair with the clock and the sign in **Appendix E.2, Figure 17 (Row 3, Column 4)**: Conditioning on the vision token containing the letters “T” and part of “O” reveals a strong interaction with another vision token of this word in the sign and the text token “town”, which suggests *the presence of a third-order interaction between these tokens*.
>
> > **W3:** It would be interesting to see how FlixLIP compared to previous works given the same budget.
>
> Methods from previous works have no budget since they do not use model inferences to explain the similarity prediction. We compared FIxLIP to Shapley and Banzhaf values given the same budget in all experiments.
>
> > **Q1:** What is WLS in Figure 1?
>
> It means a regression-based approximation with weighted least squares (WLS), which we will add to the caption.
>
> > **Q2:** How does the subset selection affect performance (Sec. 3.3)?
>
> This is a great question. We only use the subset selection for insertion/deletion evaluation and visualizing explanation graphs. While the greedy subset selection is not optimal, we already achieve *state-of-the-art faithfulness performance*, where optimizing weighted clique finding could further improve our results. Regarding efficiency performance, it takes about two seconds to find subsets of all sizes for one explanation instance. Having said that, *efficiently detecting* and *analyzing* the subsets (explanation graphs) is a very interesting line of future work.
>
> > **Q3:** What are the computational overhead for different p settings in Table 1? Also what is the budget for other methods.
>
> To clarify, there is no computational overhead for different $p$ settings. Shapley values use the model-agnostic estimator with a budget of $2^{17}$, while other methods have no budget hyperparameter (number of sampled masks) since they do not perform model inferences.

---

> > ### Comment · Reviewer_Raws · 2025-08-05
> >
> > I have carefully read authors response and updated my ratings correspondingly. Thank authors for detailed responses. It addressed most of my concerns. My only remaining concern/suggestion is adding adequate comparison to previous methods under different settings of budgets. The budgets can be different such as weights, intermediate features, or other ``budget'' that can be measured.

---

> > > ### Author Response · Authors · 2025-08-05
> > >
> > > We would like to sincerely thank the reviewer for acknowledging the quality and significance of our work, engaging in the discussion, and giving valuable suggestions.

---

### Official Review · Reviewer_gzAn · 2025-07-03

**Clarity:** 1
**Significance:** 3
**Originality:** 2
**Rating:** 4
**Confidence:** 2

**Summary:**

This paper presents an explanation method for image-text dual encoder models that is based on interactions between image patch features and text token features. In practice, the method works by sampling many masks for the image patch and text token positions, computing for each match the image-text similarity according to the model, and fitting a least-squares regression that predicts the image-text similarity from the weighted sum of interactions between pairs of image and text features. The method is evaluated in several ways: an insertion/deletion setting, a pointing game setting, and a qualitative analysis. The method generally performs better than first-order attribution methods. The experiments also include an analysis of how the computational requirements of the method scale with the number of masks sampled.

**Questions:**

- Can you explain the method with less notation?
- In the insertion/deletion evaluation, the proposed method is not uniformly better than the Shapley baseline for all choices of p. Are there hyperparams that could be tuned in the Shapley case and if so, is the reported Shapley line the best of those?

**Ethical Concerns:**

["NO or VERY MINOR ethics concerns only"]

**Limitations:**

Yes

**Quality:**

3

**Strengths And Weaknesses:**

Strengths:
- The experiments section is thorough. There are several applications or ways of evaluating the method, and confidence intervals/error bars are provided for at least some of the quantitative evaluations. In addition, a running-time analysis is provided.
- The method does provide value over first-order attribution methods according to the insertion/deletion and pointing-game evaluations.
Weaknesses:
- Due to the heavy use of notation, the method section is quite challenging to understand and is overwhelming.
- Isn’t Definition 2 circular, since we use b_i to define b?
- What does it mean for a mask to be high or low similarity (line 202)?
- Explanation of the insertion/deletion evaluation could be dramatically improved
- Line 308 needs ref (“consistent with prior work”)

---

> ### Author Rebuttal · Authors · 2025-07-29
>
> **We sincerely thank the reviewer for their time and effort in reviewing our paper.**
>
> > **Strengths:** The experiments section is thorough. There are several applications or ways of evaluating the method, and confidence intervals/error bars are provided for at least some of the quantitative evaluations. In addition, a running-time analysis is provided. The method does provide value over first-order attribution methods according to the insertion/deletion and pointing-game evaluations.
>
> Thank you for highlighting **the thoroughness of our experiments**!
>
> > **W1:** Due to the heavy use of notation, the method section is quite challenging to understand and is overwhelming.
>
> Thank you for your valuable feedback. To address it, we will add a notation table describing each symbol at the beginning of the Appendix. Our method is rooted in game theory, where we introduce the necessary mathematical notation to prove **Theorems 1 & 2**. In that, we extend the notation used in related work on applying Shapley and Banzhaf interactions in machine learning [16, 28, 44, 58].
>
> > **W2:** Explanation of the insertion/deletion evaluation could be dramatically improved
>
> To make the insertion/deletion evaluation metric clearer, we provide its comprehensive description with a visual example aiding such an explanation in **Appendix B.2, Figure 7**.
>
> > **Q1:** Isn’t Definition 2 circular, since we use b_i to define b?
>
> **Definition 2** is not circular and introduces the operator $\oplus_{M_o}$ that requires two inputs $x$ and $b$. We consider $b$ as a vector $b \in \mathbb{R}^n$, while we use its values indexed with $b_i$ to define the masking operator applied to $x$ and $b$.
>
> > **Q2:** What does it mean for a mask to be high or low similarity (line 202)?
>
> Thank you for pointing out this mental shortcut; we mean token subsets (masks) of high and low similarity as evaluated by the game (model).
>
> > **Q3:** Line 308 needs ref (“consistent with prior work”)
>
> Thank you for this valuable suggestion; we will add references [8, 13] to this sentence.
>
> > **Q4:** Can you explain the method with less notation?
>
> Our method can be described as a 3-step algorithm demonstrated visually in **Figure 1**.
>
> **Input:** The method for explaining a similarity value predicted by a vision–language encoder (e.g. CLIP) takes an image–text pair as input. It also takes a float probability parameter $p \in (0,1)$ and an integer budget parameter $m^2 \in \mathbb{N}$.
>
> 1. We sample uniformly $m^2$ variations of the input image–text pair (**Definition 2**). Each mask is sampled uniformly with probability $p$ for each token in a mask (**Section 3.1, line 126+**), and we take all pairwise combinations between $m$ masked images and $m$ masked texts (**Section 3.2,  line 155+**).
> 2. We apply the explained model to predict similarity values for each of the $m^2$ input variations (**Definition 3**).
> 3. We apply a regression-based least squares approximator to predict the model’s predictions from sampled binary masks (**Section 3.1, lines 146+**).
>
> **Output:** An explanation assigning attribution scores to each token and interaction scores to each pair of tokens in an input image–text pair (**Definition 1**).
>
> We can add such an algorithm in the Appendix or in the paper, given additional space.
>
> > **Q5:** In the insertion/deletion evaluation, the proposed method is not uniformly better than the Shapley baseline for all choices of p. Are there hyperparams that could be tuned in the Shapley case and if so, is the reported Shapley line the best of those?
>
> This is a great question. In the Shapley case, there is no such hyperparameter, while we report a line for the ‘best’ Shapley weights. The goal of our method (FIxLIP) is to allow for arbitrary sparseness of sampled masks, which is not possible in the Shapley case (see **Remark 2**). We introduce a flexible framework where the parameter $p$ can be used to achieve the best $p$-faithfulness metric results. For example, FIxLIP with $p=0.7$ outperforms FIxLIP (Shapley interactions, SI) as measured with 0.7-faithfulness correlation in **Figure 3 (left)**. Analogously, in **Figure 3 (right)**, FIxLIP with $p=0.5$ outperforms FIxLIP (SI) based on $R^2$ with $p=0.5$. In **Appendix D, Figure 10**, we show another interesting scenario analysing the SigLIP-2 (ViT-B/32) model, where FIxLIP with $p=0.4$ and $p=0.5$ outperforms FIxLIP (SI).

---

### Author Response · Authors · 2025-08-07

Dear Reviewers,

Thank you for your time and effort in reviewing our paper.

We are excited about the appreciation of our valuable method (Reviewers gzAn, Raws, WGoS), grounded in a strong theoretical formulation (Reviewers Raws, WGoS), demonstrating state-of-the-art results (Reviewer Raws) across thorough experiments (Reviewer gzAn). We are looking forward to incorporating the requested minor changes to our final revision, making it clearer for the reader.

Warm regards,

Paper5064 Authors

---

### Note · Authors · 2025-08-11

Dear Reviewers and Area Chair,

As vision–language encoders are increasingly deployed in real-world applications, ensuring their predictions are explainable becomes pivotal. To this end, we introduced faithful interaction explanations of CLIP-like models (FIxLIP), offering a unique perspective on interpreting image–text similarity predictions. Moreover, we derived three evaluation criteria to facilitate future work in this direction. FIxLIP is grounded in game theory, where we analyze its interesting properties compared to prior art. Empirically, it achieves state-of-the-art faithfulness performance in comprehensive experiments. We found FIxLIP explanations useful in visually interpreting the model’s output function, leading to intriguing discoveries. Furthermore, we use it to compare different vision–language encoder architectures, like version 1 and version 2 of SigLIP.

In brief, we addressed the 14 questions and 5 potential weaknesses across the 3 reviews as follows:
- Reviewer gzAn / W1: Added a table with mathematical notation in Appendix A.
- R gzAn / W2: Extended the description of insertion/deletion evaluation in Appendix B.2.
- R gzAn / Q1: Clarified.
- R gzAn / Q2: Fixed an unclear wording in the sentence (line 202).
- R gzAn / Q3: Added references to the sentence (line 308).
- R gzAn / Q4: Extended introduction to include a plain summary of our method.
- R gzAn / Q5: Clarified, pointing to the appropriate content in the paper.
- R Raws / W1, W2, W3: Clarified, pointing to the appropriate content, and agreed that these are interesting future work directions.
- R Raws / Q1: Added “with weighted least squares (WLS)” to the caption of Figure 1.
- R Raws / Q2: Answered.
- R Raws / Q3: Added a sentence clarifying that changing $p$ has no computational overhead.
- R WGoS / Q1: Clarified, pointing to the appropriate references.
- R WGoS / Q2: Extended the introduction to clarify “out-of-distribution”.
- R WGoS / Q3: Clarified, pointing to the appropriate content in the paper.
- R WGoS / Q4: Extended the description of Figure 1.
- R WGoS / Q5: Extended the description of Figure 2, clarifying “negative deletion values”. Extended Appendix B.2.
- R WGoS / Q6: Extended the description of the pointing game evaluation in Section 4 and Appendix B.3.

Thank you again for the valuable feedback, which allowed us to improve the manuscript.

---

### Decision · Program_Chairs · 2025-09-17

**Decision:**

Accept (poster)

**Comment:**

The paper proposes a novel approach for interpretability of CLIP-style image-text tokens, that gives insight into the cross-modal interactions between image-patch tokens and text tokens (called “second-order” in the paper).

All reviewers point out the novelty of the proposed approach and commend the thoroughness of both the technical analysis of the method, and the empirical evaluation. All reviewers recommend (weak) acceptance of the submission, and the authors were able to clear up all questions during rebuttal.

It should be noted that all reviewers provided relatively low confidence values for their reviews. Judging from the reviews themselves, this is in part because of challenges in understanding the writing of the paper. While all reviewers seem to have grasped the proposed approach in the end, multiple reviewers pointed out that reformulating the approach section of the paper could go a long way in making it easier to digest / understand. Thus, while I think that the technical novelty and thoroughness of the paper warrants its acceptance, I strongly encourage the authors to consider rewriting their approach section, and providing more intuitive explanations in addition to precise notation, which could go a long way towards strengthening the impact of the paper.